# Harnessing the Power of NK Cell Receptor Engineering as a New Prospect in Cancer Immunotherapy

**DOI:** 10.3390/pharmaceutics16091143

**Published:** 2024-08-29

**Authors:** Stefania Douka, Vasilis Papamoschou, Monica Raimo, Enrico Mastrobattista, Massimiliano Caiazzo

**Affiliations:** 1Pharmaceutics Division, Faculty of Science, Utrecht Institute for Pharmaceutical Sciences (UIPS), Utrecht University, Universiteitsweg 99, 3584 CG Utrecht, The Netherlands; 2Glycostem Therapeutics B.V., Kloosterstraat 9, 5349 AB Oss, The Netherlands; monica@glycostem.com; 3Department of Molecular Medicine and Medical Biotechnology, University of Naples “Federico II”, Via Pansini 5, 80131 Naples, Italy

**Keywords:** cancer immunotherapy, adoptive cell transfer, NK cells, tumor microenvironment, receptor engineering, NK cell clinical trials

## Abstract

Natural killer (NK) cells have recently gained popularity as an alternative for cancer immunotherapy. Adoptive cell transfer employing NK cells offers a safer therapeutic option compared to T-cell-based therapies, due to their significantly lower toxicity and the availability of diverse autologous and allogeneic NK cell sources. However, several challenges are associated with NK cell therapies, including limited in vivo persistence, the immunosuppressive and hostile tumor microenvironment (TME), and the lack of effective treatments for solid tumors. To address these limitations, the modification of NK cells to stably produce cytokines has been proposed as a strategy to enhance their persistence and proliferation. Additionally, the overexpression of activating receptors and the blockade of inhibitory receptors can restore the NK cell functions hindered by the TME. To further improve tumor infiltration and the elimination of solid tumors, innovative approaches focusing on the enhancement of NK cell chemotaxis through the overexpression of chemotactic receptors have been introduced. This review highlights the latest advancements in preclinical and clinical studies investigating the engineering of activating, inhibitory, and chemotactic NK cell receptors; discusses recent progress in cytokine manipulation; and explores the potential of combining the chimeric antigen receptor (CAR) technology with NK cell receptors engineering.

## 1. Introduction

The field of cancer immunotherapy has shown exponential growth over the last decades. Several immunotherapy methods have been developed, such as cancer vaccines, oncolytic virus therapies, cytokine therapies, immune checkpoint inhibitors (ICIs), and adoptive cell transfer therapies (ACTs) [1]. In ACT, immune effector cells are ex vivo expanded and later intravenously infused in cancer patients to enhance their immune system [2,3]. Immune T cells used for ACT are mostly autologous, i.e., sourced directly from the patient, but other types of immune cells, such as γδ Τ cells and natural killer (NK) cells, can also be allogeneic, i.e., derived from donors [1,4]. Recent developments in cancer immunotherapy with T cells include T cell receptor (TCR) engineering and the chimeric antigen receptor (CAR) T cell technology [4]. CAR–T cells have already been applied for the treatment of B-cell malignancies, multiple myeloma (MM), acute lymphocytic leukemia (ALL), and other hematological tumors [5]. Nevertheless, CAR–T cell therapies show toxic side effects, such as the possible induction of the cytokine release syndrome (CRS) or the immune effector cell-associated neurotoxicity syndrome [6]. Moreover, the usage of allogeneic T cells poses the risk of inducing graft-versus-host disease (GvHD) [6]. Thus, recent research has shifted its focus to using NK cells as a seemingly safer and more efficient alternative for cancer immunotherapy [2,6].

In contrast to T cells, NK cells offer a broader application for anti-tumor immunotherapy, mainly due to their innate cytotoxic potential [7,8]. NK cells eliminate aberrant cells without the need for pre-activation or major histocompatibility complex class I (MHC-I)-dependent antigen presentation by the targeted cells [3,9,10]. Tumor cell killing is achieved by multiple mechanisms: degranulation, i.e., the release of the cytolytic granules containing perforin and granzyme; antibody-dependent cellular cytotoxicity (ADCC), ignited by the CD16 receptor-mediated identification of antibody-coated target cells; and the induction of tumor cell apoptosis by expressing tumor necrosis factor α (TNF-α), FasL, or TNF-related apoptosis-inducing ligands [3,7,8,11,12,13]. Clinical application of allogeneic NK cell transplantation is safer compared to allogeneic T cells, as they do not induce neurotoxicity, GvHD, or CRS [2,14]. Cytokines secreted by NK cells, like IFN-γ and GM-CSF, are safer than the ones produced by T cells, such as TNF-α and IL-6 [15,16]. Finally, yet importantly, the short life span of allogeneic NK cells limits the possibility of off-target in vivo events in contrast to the longer survival of autologous T cells [8,17]. 

NK cells are defined as CD56^+^CD3^-^ large granular lymphoid cells capable of producing IFN-γ [16]. Constituting almost 10% of the peripheral blood mononuclear cells (PBMCs), NK cells are distinguished into two major subpopulations based on the expression of the low-affinity Fc gamma receptor 3A (FcγRIIIa), also known as CD16, and the adhesion molecule CD56 (NCAM); the CD56^bright^CD16^dim^ and the CD56^dim^CD16^bright^ cells. The CD56^bright^CD16^dim^ subpopulation (10% of the total PB-NK cells) consists of immunomodulatory cytokine-producing NK cells, while the CD56^dim^CD16^bright^ cells (90% of the total PB-NK cells) have stronger cytotoxicity [8,10,13].

NK cell cytotoxicity depends on their capability of identifying malignant or infected cells through their wide variety of activating and inhibitory surface receptors [10,16]. The spontaneous cytolytic killing activity of the NK cells is ignited by their activating receptors, such as NKG2D, CD16, and the natural cytotoxicity receptors (NCRs) that empower the NK cells to recognize cells with increased levels of ligands induced by stress, a state called “induced self” [14,18]. Inhibitory receptors play a pivotal role in regulating NK cell function and serve as checkpoints that control immune responses and inflammation [19]. They are often upregulated on chronically stimulated NK cells, leading to reduced activation and diminished overall response to cancer [20]. NK cells can distinguish healthy cells from malignant cells depending on the deficiency of MHC-I, also known as human leukocyte antigen (HLA) molecules, on the target cell surface. This “missing-self” state is an escape mechanism of tumor cells against T-cell-mediated killing [7]. However, it induces the exact opposite outcome against the HLA-binding NK cell inhibitory receptors, such as killer-cell immunoglobulin-like receptors (KIRs) and NKG2A/CD94, resulting in subsequent NK cell activation and killing of the cancer cells [7,14,18]. Additionally, NK cells are equipped with a variety of chemotactic surface receptors, which are responsible for their migration to chemokine-expressing tumor sites [12,21].

NK cells can be obtained from several sources, either autologous or allogeneic [2,4]. Primary NK cells deriving from peripheral blood (PB-NK), umbilical cord blood (UCB-NK), the bone marrow, or the placenta (obtained by apheresis) are the most studied sources with many applications [4,13,22]. However, due to the possible induction of GvHD by the remaining alloreactive T cells, the dose limitations of allogeneic NK cells indicate the importance of prior thorough purification of the cells [22]. In addition, hematopoietic stem/progenitor cells, human embryonic stem cells, and induced pluripotent stem cells (iPSCs) offer an off-the-shelf solution [3,23,24]. A subcategory of NK cells available for adoptive cell therapy are the allogeneic cytokine-induced memory-like (CIML) NK cells. CIML NK cells are PB-NK cells incubated in vitro with the cytokines IL-12, IL-15, and IL-18 prior to their infusion in the patient to boost their activity and persistence for weeks to months [3,13,21,24]. NK cell lines have also been explored, namely NK-92, NK-YS, KHYG-1, NKG, IMC-1, NKL, and NK3.3 [9,13]. The immortalized NK-92 cell line, deriving from a large granular lymphocyte (LGL) lymphoma patient, is currently used in clinical trials due to its indefinite proliferation, resulting in cell abundancy and easier manipulation for therapeutic purposes [23,25]. The other existing NK cell lines do not appear to have the same cytotoxic potential as the NK-92 cell line [9,13].

The NK-92 cells express only the NKG2A inhibitory receptor and not the KIR receptors, in addition to being CD16^-^ and thus unable to induce the ADCC pathway [14,23]. Furthermore, irradiation of the NK-92 cells is required to avoid oncogenic adverse events in the patients, leading to reduced cell fitness and the need for multiple infusions [8,13,14]. However, their IL-2 dependency raises toxicity issues from the possible repeated IL-2 administration [8]. For this reason, the NK-92MI cell line presents a promising alternative, as it derives from the non-viral transfection of NK-92 cells to enable endogenous production of IL-2, thereby eliminating the need for additional cytokine injections [13].

Prior to infusion, NK cells are primed and stimulated via cytokines such as IL-2, IL-12, IL-15, IL-18, IL-21, and type 1 interferons like IFN-α [26]. However, prolonged exposure to high doses of those cytokines, especially IL-2, can result in toxicity and the expansion of unwanted immunosuppressive cells, such as T regulatory cells (Tregs) [6,8]. Feeder cell lines, like the modified K562 leukemia cells expressing IL-15 or the recent version expressing IL-21 and 4-1BBL, offer an alternative strategy for ex vivo expansion and are considered safer for clinical application [21,27].

The tumor-cell-targeted cytotoxicity of NK cells is dependent on their surface receptor repertoire, other inhibitory molecules, metabolic and transcriptional factors, and the immunosuppressive tumor microenvironment (TME) [1,12,18,21]. These parameters can hinder activating signals, tumor infiltration, and NK cell proliferation. In addition, tumor immune escape mechanisms can disrupt the NK cell receptor balance and diminish the expression of the activating receptors while boosting the expression of inhibitory molecules [28]. To address this, specific monoclonal antibodies (mAbs), also known as ICIs, are employed to block the inhibitory receptors’ functions by obstructing the receptor–ligand inhibitory effect. Several mAbs targeting NK cell inhibitory receptors are currently being evaluated in clinical trials [1]. Moreover, bi- and tri-specific killer-cell engagers (BiKEs and TriKEs), also referred to as NK cell engagers (NKCEs), have been recently introduced to link NK cell receptors with tumor-presenting antigens, thus bringing NK cells and tumors into proximity and offering a cost-effective solution for NK-cell-based therapies [29].

In addition to ICIs and NKCEs, NK cells can be engineered ex vivo, thereby boosting their post-infusion persistence and cytotoxic potential [4]. One example is the recent application of CAR technology on NK cells (CAR–NK), leading to a more potent lysis of tumor cells compared to CAR–T cells, via both CAR-dependent and CAR-independent activation [3,15]. Researchers have successfully genetically manipulated NK cells via viral and non-viral methods, with many studies having already reached the stage of clinical trials [9]. Viral transduction of NK cells is possible with the use of retroviruses or lentiviruses, with safety concerns surrounding the genomic integration of viral DNA [13,30]. On the contrary, transfection of NK cells via mRNA electroporation has shown high transfection efficiency, yet it results in transient gene expression only lasts for a few days [18,31,32]. DNA transposon delivery is a rising alternative for a more persistent DNA expression of longer DNA sequences [2,9,33]. Other non-viral methods of NK cell transfection, such as nucleofection, lipofection, mechanoporation, trogocytosis, and polymer or lipid-based nanoparticles (LNPs), have also been introduced [8,30,32,34]. In particular, LNPs have obtained clinical approval for the production of mRNA vaccines against COVID-19 and have recently shown promising efficiency of RNA delivery to NK cells [35,36]. Genetic engineering of NK cells is also feasible with gene-editing techniques, such as the clustered regularly interspaced short palindromic repeats (CRISPR)/Cas9 complex, zinc finger nucleases, or transcriptional activator-like effector nucleases [8]. Figure 1 summarizes and illustrates all preclinical and clinical efforts aimed to enhance NK cell cytotoxicity.

This review provides a comprehensive overview of the NK cell receptors that are mainly used as NK cell immunotherapy targets, with a focus on the ongoing preclinical and clinical trials involving the manipulation of activating receptors, inhibitory receptors, and other pivotal factors of NK cells for monotherapy or combinational approaches in cancer immunotherapy.

## 2. NK Cell Receptor Engineering

The upcoming sections of this review will focus on the most extensively researched NK cell receptors for cancer immunotherapy, as indicated in Table 1. The expression of activating receptors is often downregulated under the influence of the TME; thus, restoring or enhancing their expression can improve tumor cell targeting [18]. Increased expression is achieved by receptor delivery either in their endogenous form or as CAR molecules. Conversely, inhibitory receptors that are upregulated in tumor sites can be blocked with ICIs or knocked out via receptor engineering. The most recent preclinical and clinical trials with NK cell receptor engineering for the field of cancer immunotherapy are summarized in Table 2 and Table 3, respectively, and are reviewed below. Table 2 does not include combinational studies with multiple antibodies or feeder cell-mediated overexpression of the most studied receptors.

### 2.1. Activating NK Cell Receptors

#### 2.1.1. CD16

The CD16a isoform of the CD16 receptor (FcRγIII), expressed by NK cells and other immune cells, induces the ADCC pathway by binding multivalently to the Fc portion of IgG, which is bound to tumor-associated antigens on the surface of tumor cells [25,57]. This leads to the intracellular phosphorylation of immunoreceptor tyrosine-based motif (ITAM) molecules of the homo- or heterodimers of CD3ζ and FcεRIγ [58], leading to the release of granzyme B and perforin as well as the generation of inflammatory cytokines directed towards the target cell (Figure 2) [59]. Surface expression of CD16 is regulated by a disintegrin and metalloprotease 17 (ADAM17), which acts as a regulatory checkpoint by causing ectodomain shedding and NK cell detachment from the tumor cell [25,60]. Such a mechanism boosts serial engagement of target cells, improves motility, survival, and targeted-cell engagement, but could limit cytotoxic activity by CD16 downregulation [61]. To overcome ADAM17 cleavage, researchers have developed a high-affinity (F158V) and non-cleavable (S197P) CD16a variant, also known as hnCD16, by introducing point mutations on the extracellular domain [61].

Successful transduction or electroporation of the CD16a gene in NK-92 cells has been achieved by different groups [12,25]. Jochems and colleagues engineered NK-92 cells via electroporation with a plasmid DNA containing both the high-affinity variant of CD16 (CD16-F158V) and IL-2, leading to increased cytotoxicity against breast and lung tumor cell lines when combined with avelumab in vitro [62]. CRISPR/Cas9 technology has been applied by Pomeroy and colleagues to knock out the ADAM17 cleavage sequence in PB-NK cells, which led to upregulation of IFN-γ production and ADCC in a PD-1 knockout combined in vitro study including tumor-targeting mAbs [58,63]. Zhu and colleagues recently engineered iPSCs with lentiviral particles expressing hnCD16. This modification led to the production of human-induced pluripotent-stem-cell-derived NK (hnCD16-iNK) cells which were more effective against both in vivo B-cell lymphoma and ovarian cancer xenograft models when combined with anti-CD20 and anti-HER2 mAbs, respectively [60]. In another study, van Hauten and colleagues successfully transduced NK cells derived from CD34^+^ hematopoietic stem and progenitor cells with hnCD16, leading to enhanced ADCC in combination with the tumor-targeting antibody rituximab when tested in vitro towards tumor cell lines and primary leukemia cells [64].

Daratumumab is a clinically effective monoclonal antibody that targets CD38 on the surface of multiple myeloma cells, exhibiting efficacy both as monotherapy and in combination with other anticancer treatments [65]. However, CD38 is also highly expressed on the surface of PB-NK cells, playing a crucial role in mediating cell activation [66]. Consequently, treatment with daratumumab has been shown to deplete NK cells in both humans and murine models, thereby diminishing the antibody’s effectiveness in ADCC [23]. To address this, ex vivo-expanded NK cells derived from an iPSC line were genetically modified via CRISPR/Cas9 and mRNA electroporation to knock out the CD38 receptor and simultaneously overexpress hnCD16, to enhance CD38-directed cytotoxicity and to induce robust ADCC in antibody-combined therapies. According to their results, CD38KOCD16^+^ NK cells exhibited significantly increased cytotoxicity in vitro against NCI-H929 myeloma cells when combined with daratumumab, reaching approximately 90% cell lysis at the highest effector/target (E/T) ratio. Conversely, CD38WTCD16^+^ NK cells and non-engineered NK cells both resulted in 60% cell lysis in combination with the mAb at the same E/T ratio [65]. A phase 1 clinical study was recently conducted to evaluate the dose tolerability of CD38KOCD16^+^ NK cells (FT538) in combination with daratumumab in six MM patients [clinicaltrials.gov ID: NCT04614636]. Both doses tested (1 × 10^8^ and 3 × 10^8^ cells) were well-tolerated, with no signs of neurotoxicity, CRS, or GvHD [67]. Additionally, the safety and efficacy of the FT538 and daratumumab combination were assessed in another phase 1 clinical study involving AML patients, with doses ranging from 1 × 10^8^ to 15 × 10^8^ cells [clinicaltrials.gov ID: NCT04714372]. Two out of four patients achieved an objective response rate of 50% and subsequently received an additional hematopoietic stem cell transplant, which further prolonged their survival. Although no neurotoxicity or dose-limiting toxicity were observed, 80% of the patients experienced infection and febrile neutropenia, and one patient exhibited CRS symptoms [68].

Bispecific antibodies and innate cell engagers (ICE) bring together the target tumor cells and NK cells by simultaneously binding tumor antigens and engaging with CD16 [69]. In a study conducted by Toffoli and colleagues, a bispecific single domain antibody (VHH) was constructed to target C21 for CD16 and 7D12 for epidermal growth factor receptor (EGFR) [57]. The application of this engager resulted in in vitro and ex vivo augmented activation of patient PBMC NK cells and lysis of EGFR-expressing tumor cell lines and metastatic colorectal patient-derived cancer cells [57]. The construction of another BiKE construct, which acts through CD16 while also targeting the myeloid differentiation antigen CD33, could induce the degranulation and lytic ability of NK cells towards acute myeloid leukemia (AML) cells in vitro [15,70]. In another study by Sarhan and colleagues, a TriKE construct was developed to target myelodysplastic syndrome, which engages CD16 and CD33 on the surface of myelodysplastic cells and simultaneously induces the self-production of IL-15 [71]. The engager contains two distinct scFv domains that recognize CD16 and CD33, respectively, and are connected by a linker encoding for IL-15, which binds to the IL-15 receptor on NK cells, thereby enhancing NK cell activity [72,73]. Finally, Kerbauy and colleagues successfully targeted CD30^+^ tumor cells with an AFM13 BiKE construct utilized for complex CD30 expressed by lymphoma or leukemia cells with CD16^+^CB^-^NK cells while testing the influence of prior NK cell activation with IL-15 in vitro and in vivo [24,74]. The AFM13 BiKE is currently being tested in the clinic in combination with non-engineered allogeneic NK cells and AFM13-pretreated CB-NK cells (modified NK cells) for CD30^+^ Hodgkin lymphoma and non-Hodgkin lymphoma [clinicaltrials.gov ID: NCT05883449 and NCT04074746]. Previous clinical studies from Affimed GmbH assessing AFM13 safety and tolerability in patients with Hodgkin lymphoma demonstrated activation of NK cells and a reduction in CD30 levels in peripheral blood samples [clinicaltrials.gov ID: NCT01221571]. Moreover, 61.5% of evaluated patients showed overall disease control, including 11.5% who demonstrated partial remission. When a higher AFM13 dose was administrated (≥1.5 mg/kg), the disease control rate was 77% [75].

#### 2.1.2. NKG2D

The natural killer group 2 member D protein (NKG2D) activating receptor belongs to the C-lectin family and can recognize ligands associated with viral- or bacterial-infected cells, but most notably with tumor-transformed cells [16,76]. NKG2D plays an important role in the anti-tumor activity of NK cells due to its ability to bind to MHC-I-chain-related molecules, such as MICA, MICB, and the UL16-binding proteins (ULBPs) [10,16,37]. In mice, the retinoic acid early inducible-1 gene (RAE-1) and the UL16-binding protein-like (MULT)-1 have also been reported as NKG2D ligands [16]. These NKG2D ligands are overexpressed by tumor-transformed cells and infected cells, thus distinguishing them from healthy cells [37,76]. NKG2D induces NK cell cytotoxic activity and cytokine secretion via signaling through the DNAX-activating protein of 10 kDa (DAP10) in humans and DAP12 in mice (Figure 2) [37,76].

Malignant cells often escape immune cell targeting by secreting soluble forms of the NKG2D ligands, decreasing the effectiveness of NKG2D signaling [76]. Analyzing the specific molecular pathways of NKG2D downregulation in tumors, Xing et al. reported that both soluble and surface forms of MICA and MICB cause the desensitization of NK cells [77]. Other studies have shown that histone deacetylases (HDACs) inhibitors can induce higher levels of MICA/MICB expression on tumor cells, thus promoting NK cell tumor activity [77]. MICA-gene-specific transcriptional activation and overexpression by tumor cells using CRISPR/Cas9 technology were studied by Sekiba and colleagues and resulted in increased NKG2D-mediated clearance of the targeted cells in vitro [78,79]. Chitosan-based nanoparticles were employed by Tan and colleagues for the successful in vivo delivery of a plasmid encoding NKG2D and IL-21 (dsNKG2D-IL-21) into CT-26-induced solid tumors in mice, inducing the augmented secretion of the ligand and cytokines [80]. This led not only to NK cell but also to T cell stimulation and migration to the tumor tissue [80,81]. In another study by Youness and colleagues, overexpression of miR-486-5p in primary NK cells through lipofection caused an increase in NKG2D and perforin in vitro [82]. In addition, the NK-92 cell line was lentivirally transduced by Sayitoglu and colleagues to overexpress the NKG2D receptor to target sarcomas. NKG2D^+^ NK-92 cells showed enhanced degranulation towards all sarcoma explants and all tested tumor cell lines apart from the neuroblastoma cell line SH-SY5Y [83]. Poly ADP-ribose polymerase 1 (PARP1) is known to repress NKG2D ligand expression, thereby promoting immune escape. Its inhibition using either siRNA methodologies or inhibitors like talazoparib, olaparib, and AG-14361 has been shown to significantly increase ligand levels on AML stem cells by 6- to 50-fold [84]. The German Cancer Research Center is now initiating a clinical trial for AML, combining allogeneic NK cell therapy with the PARP1 inhibitor talazoparib to induce re-expression of NKG2D ligands on tumor cells [clinicaltrials.gov ID: NCT05319249].

The knockout of NKG2D has also been studied by several groups, with findings proving its importance in cancer immunotherapy. Inhibition of NKG2D in early NK cell developmental stages has been associated with the hyperactivity of the NKp46 (NCR1)-activating receptor and the targeting of NKp46-ligand-expressing tumors [85]. Wang and colleagues co-incubated the Kasumi-1 AML cancer cell line with NK92MI cells and the anti-NKG2D antibody, resulting in a reduction in the apoptotic ratio of the AML cell line, demonstrating the importance of the NKG2D cytotoxicity potential [76]. In another study, a BiKE construct was produced combining an anti-CS1 scFv domain and an anti-NKG2D scFv domain [15]. Its application in an in vitro human MM model proved the dose-dependency of IL-2-primed PBMC-derived NK cell cytotoxicity and cytokine secretion in the presence of the NKG2D receptor [86]. Similarly, a BiKE construct with Fab fragments for the binding of NKG2D and HER2, a tumor-expressed antigen, could stimulate NK cell cytotoxicity in vitro [87,88]. Novel strategies with bi-specific immunoconjugate constructs are focused on simultaneous targeting of NKG2D ligands, like MICA or ULBP1/2, and tumor-expressed antigens such as BCMA, CD19, and VEGFR2 [87]. In vivo preclinical studies confirmed the efficiency of such molecules in inducing NKG2D activation in NK cells, with subsequent enhancement of NK cell cytotoxic activity towards the respective antigen-presenting tumor cells [87].

#### 2.1.3. NKG2C

The C-type lectin CD94/NKG2C-activating receptor is highly expressed by the adaptive CD56^dim^ NK cells, usually post-stimulation by a cytomegalovirus (CMV) infection, and is very potent for ADCC induction and IFN-γ secretion [38,89]. Incubation of allogeneic NK cells with feeder cells and IL-15 induces NKG2C^+^ adaptive NK cells with cytotoxic potential [45]. Haroun-Izquierdo and colleagues produced adaptive single self-KIR^+^NKG2C^+^ NK cells, named ADAPT-NK cells, which have a higher proliferation rate compared to common adaptive NK cells, in both in vitro and in vivo AML tumor cell models [38]. The improved expansion and activity of these cells were caused by three factors: (1) the expression of single self-KIR, which provided higher alloreactivity; (2) the targeting of HLA-E; and (3) the induction of CD16-mediated ADCC [38]. NKG2C^+^CD57^+^ NK cells, deriving from CMV-seropositive donors, highly expressed those molecules after CMV reactivation post-hematopoietic stem cell transplantation (HSCT), depicting a potent activity of cytolysis and a “memory-like” behavior [90]. The same memory NK cell subtype was used on bone marrow-transplanted patients with leukemia to reduce the relapse rate [91,92]. In another study, NKG2C^+^ NK cells were expanded with an engineered feeder cell line to express HLA-E*spG, an artificial disulfide-stabilized trimeric HLA-E ligand [93]. When tested against K562 and primary glioblastoma multiforme cells, these NKG2C^+^ cells showed enhanced cytotoxicity in vitro compared to wild-type NK cells. Finally, an anti-NKG2C/IL-15/anti-CD33 TriKE construct was produced by Chiu and colleagues in order to target CD33^+^ AML cells with NKG2C^+^ CMV-reactivated patient-derived PB-NK cells but also NKG2C-engineered iPSC-derived NK cells [94]. The outcome of this study was an increase in NKG2C^+^ NK cell proliferation, cytotoxicity, degranulation, ΙFN-γ production, and efficient AML tumor cell elimination in vitro [94].

#### 2.1.4. NKp46

NKp46 (NCR1) is a crucial activating receptor of the NK cells, responsible for stimulating their cytolytic activity and cytokine secretion [95]. NKp46 signals through phosphorylation of two ITAM-bearing molecules, CD3ζ and FcR-γ (Figure 2) [95]. As mentioned earlier, NK cell TriKEs combining an NKp46 scFv-binding domain, a CD16 Fc binding domain, and a tumor-associated antigen have been manufactured, leading to augmented ADCC and NK cell cytotoxicity against mouse cancer [72,87,95]. Another available multi-specific killer engager called FLEX-NK, developed by Cytovia Therapeutics in 2021, targets NKp46 and GPC3, a glycoprotein expressed by solid tumors, and CD38 to simultaneously eliminate solid tumors and MM. This engager was tested in a preclinical hepatocellular carcinoma mouse model, leading to increased NK cell detection and enhanced tumor inhibition [96].

#### 2.1.5. NKp44

NKp44 (NCR2) is only found in human-activated NK cells and other types of immune cells, and it has been associated with the identification of transformed cells, signaling via DAP12 (Figure 2) [97]. Several NKp44 ligands have been mentioned in the literature, with the most dominant being Nidogen-1, the proliferating cell nuclear antigen (PCNA), the mixed-lineage leukemia-5 protein (21spe MML5), and viral hemagglutinins [40,41,42,43]. Barrow and colleagues found that the platelet-derived growth factor isoform PDGF-DD, produced by tumor cells, is recognized by NKp44 [97]. This results in IFN-γ and TNF-a secretion by the NK cells, as shown in NCR2-transgenic mice experiments [97]. Finally, in another in vitro study on MM cells, the inhibitory binding of PCNA on cancer cells with NKp44 was blocked with the mAb 14-25-9, enhancing NK cell anti-tumor activity, IFN-γ production, and degranulation [98].

#### 2.1.6. NKp30

NKp30 (NCR3, CD337) belongs to the immunoglobulin superfamily and is a type I transmembrane NK cell receptor, signaling via the ITAM-associated molecules CD3ζ and FcεRIγ (Figure 2) [87]. The activation of NK cells through recognition of the NKp30 ligands, such as the surface molecule B7-H6 and the nuclear factor HLA-B-associated transcript 3 (BAT3)/Bcl2-associated athanogene 6 (BAG6), stimulates NK cell cytotoxicity and cytokine secretion [44,87]. NK cell killer engager molecules, also known as immunoligands, have been produced for simultaneous NKp30 and EGFR targeting, with the bi-specific construct consisting of a humanized Fab variant from the Ab cetuximab and affinity-optimized variants of the N-terminal Ig-like V-type domain of B7-H6 [44]. These BiKEs successfully stimulated EGFR-positive tumor cell killing and IFN-γ and TNF-α secretion by NK cells [44,87]. A follow-up study targeting NKp30 on NK cells and EGFR on tumor cells used a BiKE construct combining NKp30-specific VHH of Camelidae origin and the EGFR-specific humanized Fab of cetuximab, leading to a more potent NK-cell-mediated tumor-killing effect in vitro when compared to the previous B7-H6 construct or cetuximab treatment [99]. Finally, Compass Therapeutics developed CTX-8573, a multi-specific construct combining anti-NKp30 Fab fragments and the C-terminus of the B-cell maturation antigen (BCMA) [87]. It has been used to engage with CD16 and NKp30 NK cell receptors in both in vitro and in vivo studies [27,87].

### 2.2. Inhibitory NK Cell Receptors

Alongside enhancing NK cell cytotoxicity through induction of the activating receptors, downregulating the expression of inhibitory receptors can also shift the NK-signaling balance towards effective anti-tumor activation [18]. These receptors and molecules are presented in Figure 3 and can be divided into two subgroups based on their binding to HLA molecules, namely HLA-specific or non-HLA-specific.

#### 2.2.1. HLA-Specific Inhibitory Receptors

##### KIR

The killer-cell immunoglobulin-like receptor (KIR) family consists of polymorphic activating and inhibitory transmembrane proteins critical for the interaction between NK cells and major MHC-I molecules, particularly MHCIa [24,45]. Out of the fourteen KIR receptors, seven are inhibitory (KIR2DL1, KIR2DL2, KIR2DL3, KIR2DL5, KIR3DL1, KIR3DL2, and KIR3DL3), and six are activating receptors (KIR2DS1-2DS5, KIR3DS1), depending on the presence of an intracellular tail on the activating domain [21,45]. The remaining KIR2DL4 has both activating and inhibitory features [21,33,100]. KIR inhibitory receptors are important for the “missing-self” state recognition by the NK cells, facilitating the distinction between healthy and malignant cells in the absence of MHC-I on the tumor cell surface [11,45].

Anti-KIR antibodies, which block the KIR association with HLA-C molecules and the subsequent inhibition of NK cell cytotoxic activity, have recently reached the stage of clinical trials reporting limited side effects [101]. The humanized IgG4 mAb 1-7F9 (IPH2101, anti-KIR2D) was produced for the blockage of KIR2DL1/L2/L3 and KIR2DS1/S2 receptors binding to HLA-class I ligands [11,15]. In preclinical studies, this antibody induced longer NK cell survival against AML cells, and when tested in AML and MM patients, successful KIR2D binding also increased NK cell cytotoxicity [11,27,102,103,104]. IPH2101 was co-administered with lenalidomide, an agent that enhances NK cell activity by increasing the ligands of NK-cell-activating receptors, in phase I/II clinical trials with AML and MM patients [50,104,105] [clinicaltrials.gov ID: NCT01248455]. Nevertheless, further studies were suspended since lower NK cell degranulation and cytokine secretion were observed, possibly due to phagocytes removing KIR2D from the NK cell surface via FcγRI-mediated trogocytosis [21]. Lirilumab (IPH2102) is a recombinant version of IPH2101 and is the first fully clinically tested anti-KIR monoclonal antibody, created for the inhibition of the KIR–HLA class I ligand interaction in autologous NK cell applications [11,13]. The effectiveness of this anti-KIR2DL1/2/3 antibody was first confirmed by Sola and colleagues in an HLA-C-expressing B-cell lymphoma xenograft model on RAG-1-deficient mice [106]. Yet, the clinical efficacy of lirilumab monotherapy remains debatable [24,72,103]. Lirilumab has also been used in combination therapy with the immunosuppressing drug lenalidomide to induce ADCC [11].

Using different combinations of anti-PD-1, anti-CD20, anti-SLAMF7, and 5-azacytidine in ongoing clinical trials has shown promising results in KIR inhibition [101,103]. Moreover, clinical studies on advanced or metastatic solid tumor patients are currently ongoing with the combination of lirilumab, nivolumab (targeting PD-1), and ipilimumab (targeting CTLA-4) [21,106,107]. A recent addition to the anti-KIR antibody group is IPH4102 (lacutamab), a humanized IgG1 mAb targeting KIR3DL2 [102,103]. This antibody has already entered the phase of clinical studies, has shown a good safety profile for relapsed/refractory cutaneous T-cell lymphoma, and has been used in combination with chemotherapy [102,103,107]. Finally, in in vivo studies performed by Wei and colleagues, the HHLA2^+^ human lung cancer cell HCC827 was used to challenge immunodeficient mice, showing how KIR3DL3 inhibition can enhance NK cell activity [108].

##### NKG2A/CD94

One of the most important inhibitory receptors of NK cells, NKG2A, encoded by the killer-cell lectin receptor C1 (KLRC1) gene, binds to CD94 and forms a heterodimer. This complex recognizes the tumor-cell-expressed non-classical MHCIb molecules HLA-E in humans or the Qa-1 molecule in mice [45,72]. The ligation of NKG2A/CD94–HLA-E induces inhibitory signals via the phosphorylation of the tyrosine residue in the immunoreceptor tyrosine-based inhibitory motif (ITIM) domain [45]. The interaction of NKG2A/CD94 is also important for the “missing-self” recognition ability of the NK cells [45]. Notably, NKG2A/CD94 facilitates NK cell migration and contact with the target, and subsequently, the cytotoxicity and serial-killing capacity [45,109]. Due to their non-polymorphic nature, the literature has mentioned both NKG2A/CD94 and HLA-E as potential therapeutic targets to enhance NK cell anti-tumor-killing efficacy [105].

PB-NK lentiviral vector-driven RNAi knockout of the NKG2A gene was achieved by Figueiredo and colleagues, resulting in enhanced NK cell activity in vitro against an HLA-E-expressing B-lymphoblastoid cell line [12,110]. Similar results were also observed on AML-derived HLA-E-negative K562 cells, possibly due to the stimulation by the increased levels of NKp30 in the NKG2A-deficient cells [110]. Another study targeted NKG2A/CD94 using protein expression blockers (PEBLs) [18,111]. Such constructs include an anti-NKG2A antibody single-chain variable fragment connected with endoplasmic reticulum-retention domains, which block the NKG2A transport from the endoplasm to the cell membrane [18]. NKG2A-deficient NK cells had improved cytotoxic functionality in targeting HLA-E-expressing and HLA-E-deficient cancer cells deriving from Ewing’s sarcoma, osteosarcoma, and AML [12,111]. They also successfully interfered with the de novo expression of NKG2A/CD94, induced via IL-12 incubation, without any side effects on NK cell proliferation [18,111]. The in vitro results were also confirmed in in vivo xenograft models [111]. Gene-editing methods using CRISPR/Cas9 technology to knock out the killer-cell lectin receptor C1 (KLRC1), encoding for NKG2A in primary NK cells, have been applied by Bexte and colleagues. This group successfully generated a 90% NKG2D^-^ NK cell population after cell sorting, which resulted in enhanced cytotoxicity towards an HLA-E-expressing B lymphoblast cell line [112]. Similar results were obtained by Mac Donald and colleagues when tested in solid tumor cell lines in vitro and in a xenogeneic mouse model of metastatic breast cancer in vivo. Interestingly, the expression of the NKG2C-activating receptor was enhanced in KLRC1-knocked-out NK cells, a fact that also contributed to their advanced cytotoxicity [113].

Most recently, the inhibition of NKG2A/CD94 has been achieved using the humanized anti-NKG2A/CD94 mAb monalizumab (IPH2201), which has already been tested both in vitro and in vivo [103]. Monalizumab has been proven to stimulate not only NK cell activity but also CD8^+^ T effector cell anti-tumor functionality, both in mice and in humans [50,114]. NK cells from chronic lymphoid leukemia (CLL) patients showed recovery of their cytotoxic activity in preclinical studies with monalizumab [105,115]. Monalizumab has also been applied in combination therapies with other therapeutic mAbs [15,27,39]. More specifically, co-administration of monalizumab and cetuximab, an anti-EGFR antibody, in patients with head and neck squamous cell carcinoma (HNSCC) resulted in the induction of anti-tumor memory [27,39]. The co-administration with the anti-programmed death protein 1 (PD-L1) mAb durvalumab showed a similar effect for a variety of tumor types, such as mouse lymphoma and colorectal cancer (CRC) patients [50,73,114].

#### 2.2.2. Non-HLA-Specific Inhibitory Receptors

##### TIGIT

T cell immunoglobulin and immunoreceptor tyrosine-based inhibitory motif domain (TIGIT) is a non-MHC class I-specific NK cell receptor expressed by both T cells and NK cells. It negatively regulates NK cells by blocking their cytotoxic activity via competing with the DNAM-1 (CD226) receptor for the binding of its ligands, poliovirus receptor PVR (CD155), and Nectin-2 (CD112) [13,39,46,50,116]. TIGIT has a higher binding affinity towards PVR compared to DNAM-1. Therefore, the inhibition of DNAM-1 is linked to NK cell exhaustion and suppression of IFN-γ production [46,50,116]. Its inhibitory signals derive from the phosphorylation of the immunoglobulin tail tyrosine (ITT) and ITIM domains located in the cytoplasmic region [117]. Interestingly, Jia and colleagues proposed in their functional experiment studies in AML patients that TIGIT expression might be correlated with increased secretion of IFN-γ and TNF-α cytokines and granzyme B, supporting the NK anti-tumor activity in AML [105].

Recent studies have mostly focused on the effect of TIGIT inhibition on CD8^+^ T and Treg cells and its competition with DNAM-1, either with anti-TIGIT mAbs as monotherapy or in combination with anti-PD-1 and anti-PD-L1 [72,118]. Currently, there are multiple clinical trials focusing on TIGIT inhibition [119]. For instance, the TIGIT antibodies vibostolimab, etigilimab, and tiragolumab are being clinically evaluated in phase I and III clinical trials [120]. The blockade of TIGIT could be a promising alternative to enhance anti-tumor immunity. Nevertheless, the exact mechanism of action behind the NK cell interaction with CD8^+^ T cells upon TIGIT inhibition is yet to be fully defined [116,121]. 

An anti-TIGIT mouse mAb, called 13G6, was produced and used to block TIGIT activity in a CT26 colon cancer, 4T1 mammary cancer, or methylocholanthrene (MCA)-induced fibrocarcinoma mouse model [116,118,122]. Interestingly, the use of this antibody resulted in tumor growth inhibition via tumor-infiltrating NK cell protection and in the increase in CD8^+^ T cell responses and elimination of tumor cells [118,121,122]. In a more recent mouse study, B16F10 and LWT1 metastatic melanoma tumor types were eradicated by combining anti-TIGIT mAb with IL-15-induced NK cell cytotoxic activity [118,123]. Similar results were obtained with this combination approach in a B16 melanoma mouse model, where NK cell-specific TIGIT-deficiency resulted in the upregulation of DNAM-1 (CD226) expression by tumor-infiltrating NK cells and in tumor growth inhibition in CD155-deficient mice [124]. Another study with TIGIT-expressing NK cells found that inhibiting this receptor reduced the immunosuppressive effect of myeloid-derived suppressor cells (MDSCs) on NK cytotoxicity [122].

Furthermore, efforts have been made to genetically modify TIGIT in order to eliminate its inhibitory functions and, therefore, promote NK cell activity [6]. The TIGIT knockout effect was studied by Zhang and colleagues, demonstrating positive outcomes regarding NK cell protection and tumor immunity in mouse models [6,15,116]. The knockout of the TIGIT receptor from expanded NK cells using electroporation was also performed by Hasan and colleagues, demonstrating higher cytotoxicity against various cancer cell lines and spheroids in vitro [125]. Finally, an innovative approach of TIGIT exploitation came from Lupo and colleagues, who genetically engineered iPSCs-derived NK cells with lentiviral transduction to express a genetic construct called synNotch to simultaneously target CD155 and CD73 on the surface of glioblastoma cells. In short, this construct contains an extracellular TIGIT domain coupled with a synNotch transmembrane domain and the GAL4-VP64 transcription factor. Post-TIGIT binding to CD155, a conformational change yields the cleavage of a peptide and the subsequent release of the transcription factor, which promotes the generation of an anti-CD73 antibody to bind to CD73. After successful cell engineering and expansion, synNotch-iNK cells were intracranially injected in an orthotopic, patient-derived xenograft mouse model for glioblastoma. Mice treated with engineered iNK cells exhibited a significant decrease in tumor growth and enhanced survival (40% probability after 50 days) compared to those treated with non-engineered iNK cells (0% probability after 50 days). Moreover, studies on isolated brain samples showed increased levels of NKp46^+^ and granzyme B^+^ NK cells in mice treated with synNotch iNK cells, indicating an enhanced functional activation of NK cells at tumor sites. Their results also suggest effective reprogramming of the glioblastoma TME via higher CD8^+^ T cell recruitment [126].

##### CD96

CD96 is a type I transmembrane Ig glycoprotein, and even though it is mostly expressed by T cells, it has recently been indicated as another inhibitory receptor of NK cells [121]. CD96 interacts with PVR (CD155) expressed by tumor cells, similar to the interaction of TIGIT and DNAM1 [39,50]. This binding leads to a decrease in IFN-γ production by the NK cells [50]. Stimulation of human NK cells with IL-15 or TGF-β led to upregulation of CD96 expression [20]. Higher CD96^+^ NK cell levels, with distinctive exhaustion and cytokine secretion, have been found in intra-tumoral sites of hepatocellular carcinoma (HCC) patients or ovarian cancer ascites cases [20,39,50]. Additionally, it has been reported that CD96 knockout increased NK cell cytokine production and lung metastasis control in mouse models [50]. In animal models, CD96 blocking with anti-CD96 mAb has demonstrated higher efficacy when combined with anti-CTLA-4 or anti-PD-1 mAbs [127]. The impact of CD96 inhibition was not significant in terms of cytokine production and degranulation [20]. The precise mechanism of action of CD96 on NK cells, either inhibitory or activating, remains controversial [121,128].

##### CD112R

Recently identified as a lymphocyte inhibitory receptor, CD112R (or PVRIG) can ligate with CD112 with higher affinity compared to TIGIT or DNAM-1 [20,129]. It is expressed on both CD16-positive and CD16-negative NK cells, as well as by intra-tumor NK cells in prostate and endometrial cancer patients [20]. The use of anti-CD112R mAbs in a CD112^+^ and CD155^+^ human breast cancer cell line enhanced NK cell cytokine production and degranulation, confirming the inhibitory functions of CD112R [20,130]. In vivo studies performed by Li and colleagues revealed that anti-CD112R mAbs protected mice against subcutaneous MC38 tumors due to increased degranulation and IFN-γ production by tumor-infiltrating NK cells [20,131]. The combination of TIGIT and CD112R inhibition by Xu and colleagues led to enhanced NK-cell-mediated ADCC against breast cancer cells coated with trastuzumab in vitro [20,129,130,132].

##### CTLA-4

Cytotoxic T lymphocyte-associated antigen 4 (CTLA-4 or CD152) is expressed at very low levels by NK cells compared to its expression by T cells and Treg [47,49]. CTLA-4 binds with high affinity to B7-1 (CD80) and B7-2 (CD86) [49]. The expression of CTLA-4 is mainly observed in murine NK cells after cell treatment with IL-2 [20]. An FDA- and EMA-approved anti-CTLA-4 monoclonal antibody, ipilimumab, has already been characterized for its effect on T cells and is currently being clinically tested in combination with the anti-PD-1 mAb nivolumab [49]. Treatment with ipilimumab stimulated ADCC and TNF-α secretion by NK cells in vitro and resulted in tumor inhibition in a chimeric murine xenograft model [133]. CTLA-4 inhibition via treatment with ipilimumab and tremelimumab demonstrated promising results in NK cell survival rate in melanoma and malignant pleural mesothelioma patients [134]. The combination of ipilimumab with cytokine-induced memory-like (CIML) NK cell infusion and an IL-15 superagonist is currently being tested in a phase 1 clinical trial for advanced head and neck cancer [clinicaltrials.gov ID: NCT04290546]. Despite the encouraging results, the low expression of CTLA-4 in NK cells could limit its therapeutic application [47].

##### PD-1

The membrane receptor PD-1, also known as CD279, is a T-cell-associated checkpoint molecule that is also expressed by NK cells and upregulates upon CMV infection or interaction of NK cells with tumor cells [28,39]. PD-1 is bound by PD-L1 and PD-L2 to facilitate immune cell inactivation [28,39]. PD-1 is endogenously expressed by NK cells of cancer patients, and its expression can be increased by soluble factors released from the TME [39,135]. Reports have mentioned the presence of PD-1^+^ NK cells in ovarian carcinoma, Hodgkin lymphoma, intestinal adenocarcinoma, Kaposi sarcoma, bladder carcinoma, lung, breast, and uterine cancer patients [28,39].

In a promising gene-editing approach, CRISPR/Cas9 technology was used by the Moriarity group on PBNK cells to knockout the inhibitory genes encoding for PD-1 and ADAM17, performing the targeted integration through homology-directed repair using a recombinant adeno-associated virus [2,63]. Although these in vitro studies demonstrated augmented cytotoxic killing by the NK cells via non-ADCC-related pathways, in vivo results demonstrated only a modest enhancement in survival [63].

Preclinical studies in MM also showed that blocking PD-1 enhanced NK-cell-killing activity [105]. Hsu and colleagues demonstrated within their in vivo mouse studies that inhibition of PD-1 or PD-L1 can stimulate NK cell activity [39,136]. Moreover, PD-1/PD-L1 blockade seems to be crucial for HLA-negative cancer types [50]. For instance, inhibition of PD-1/PD-L1 in MHC-I-deficient tumor cells, like most Hodgkin’s lymphomas that express higher levels of PD-L1, resulted in a better response and in a higher NK cell activity in patients [50]. Makowska and colleagues have confirmed the role of IFN-β in inducing PD-L1 expression by nasopharyngeal carcinoma cells (NPC) and PD-1 expression by NK cells [137,138]. Most importantly, those studies showed that the stimulation of NK cell anti-tumor cytotoxicity by chemotherapy could be supported by PD-1/PD-L1 inhibition with anti-PD-1 antibody treatment in NPC patients [137,138]. This was confirmed in a phase 2 clinical study involving patients with late-stage non-small cell lung cancer treated with autologous NK cells and the PD-1 antibody sintilimab [clinicaltrials.gov ID: NCT03958097]. This second-line treatment demonstrated promising antitumor activity with no unexpected adverse events. Two out of twenty patients experienced neutropenia, hypertriglyceridemia, and elevated creatine kinase, which were attributed to the ICI rather than the NK cell transfer [139]. In mouse models, the combination of anti-PD-1 and anti-KIR showed the pivotal role of NK cells in targeting lymphoma tumors [140].

Several anti-PD-1 antibodies have been designed, such as nivolumab, pidilizumab, and pembrolizumab, and their effects have been mostly studied at a clinical stage for blocking T cell immunosuppression by PD-1 [15]. However, the anti-PD-1 mAb nivolumab has been used in combination therapies for solid tumors to successfully restore NK cell activity [47]. Lastly, anti-PD-1 and/or anti-LAG-3 mAbs have been used in combination with IL-12 treatment to induce NK cell stimulation in a 4T1 metastatic breast cancer mouse model [141].

##### TIM-3

T cell immunoglobulin mucin receptor 3 (TIM-3), also known as hepatitis A virus cellular receptor 2, is another inhibitory receptor of CD16-high NK cells that plays an important role in the NK cell maturation process [21,48]. The association of TIM-3 with galectin-9, expressed in various metastatic cancer types, has been reported to induce either IFN-γ production or suppression of NK cell cytotoxic activity [27,47,48]. In an in vitro study performed by Xu and colleagues, the stimulation of CXCR1 and CXCR3 chemotactic NK cell receptors following activation of the TIM-3 pathway was verified [48]. Thus, this NK cell receptor might have both activating and inhibiting effect on the NK cell anti-tumor activity [142]. Other ligands of TIM-3 that have been reported are phosphatidylserine, carcinoembryonic antigen-related cell adhesion molecule 1 (CEACAM1), and high-mobility group protein B1 (HMGB1) [142].

The expression of TIM-3 has been shown to be upregulated in the presence of TNF-α [50]. TIM-3^+^ NK cells have been identified in lung cancer and advanced-stage melanoma, showing its suitability as a target for non-small cell lung cancer and other types of anti-tumor treatment [21,134]. The inhibition of TIM-3 in vitro resulted in IFN-γ and TNF-α restoration and revitalization of NK cell cytotoxic activity in liver, lung, and melanoma cancer [20,50,134,143]. Several other preclinical studies on tumor-bearing mouse models have proven the efficacy of TIM-3 knockout or antibody blocking on the NK cell contribution in inhibiting sarcoma, colon carcinoma, or prostate tumor growth [50,105]. The combination of antibodies blocking TIM-3 and PD-1 proved to be very effective in in vivo models of melanoma, fibrosarcoma, colon cancer, and leukemia [50]. In another in vitro study on glioblastoma by Morimoto and colleagues, the knockout of TIM-3 with the use of CRISPR/Cas9 technology was shown to be beneficial in increasing inhibition of glioblastoma tumor cell growth without interfering with the expression of other NK cell inhibitory receptors [142]. Anti-TIM-3 antibodies have reached the stage of clinical trials as monotherapy or in combination therapies with anti-PD-1 and anti-LAG-3 antibodies, mostly focusing on their effect on T cells [134]. In clinical trials of patients with solid tumors, the use of a TIM-3/PD-1 bi-specific antibody is also under examination [134].

### 2.3. Chemotactic Receptors

Several NK cell receptors control chemotaxis in response to chemokines secreted by tumor cells [12]. The most important chemotactic receptors of NK cells are CXCR4, CCR7, CXCR3, CXCR2, CXCR1, CX3CR1, and CCR3-CCR5 [21,144]. Following association with their respective ligands, these receptors facilitate efficient NK cell migration and infiltration towards the tumor site, an essential process for cancer eradication [12].

#### 2.3.1. CXCR4

The importance of CXCR4 and its interaction with CXCL12 (or SDF-1a) for the homing of NK cells to the bone marrow has previously been proven by inhibiting CXCR4–SDF-1a binding [53]. Similarly, Levy and colleagues successfully transfected NK cells in vitro with CXCR4 mRNA via electroporation, suggesting that CXCR4 overexpression is a promising approach to tackle tumors in the bone marrow, such as myeloma and leukemia [53]. In studies performed by Yang and colleagues, upregulation of IL-18 expression and subsequent NK cell activation were observed after CXCR4 knockout via tissue-specific LysM-Cre-mediated recombination gene editing [54]. This method resulted in total remission and higher survival rate in mouse models [54]. Furthermore, when the NK cell line YTS was lentivirally transduced with an anti-EGFRvIII CAR construct and CXCR4, the chemotaxis towards glioblastoma cells and cancer survival in xenograft mouse models were enhanced [145,146].

#### 2.3.2. CCR7

Other studies have focused on CCR7, a receptor responsible for the lymph node homing of NK cells, aiming to improve tumor targeting by NK cells [147]. K562 feeder cells were used to transfer CCR7 to NK cells via trogocytosis, leading to increased NK cell migration to the lymph nodes in athymic nude mice via binding to the associated cytokines CCL19 and CCL21 [6,148]. Moreover, this specific homing of NK cells has been enhanced by Carlsten and colleagues via CCR7 mRNA electroporation of primary NK cells in combination with CD16 to induce ADCC and NK-cell-mediated killing of lymphoma cells in vitro [149]. Finally, the genetic modification of NK-92 cells via lentiviral transduction to simultaneously overexpress CXCR4 and CCR7 receptors on the cell surface showed increased migration towards human colon cells and ameliorated tumor prognosis in mouse xenograft models [55].

#### 2.3.3. CXCR3

Another important chemotactic NK cell receptor is CXCR3, which binds to the IFN-γ-induced and tumor-secreted CXCL9, CXCL10, and CXCL11 chemokines [23,150]. CXCR3 is crucial for NK cell solid tumor infiltration [150]. In a study using irradiated EBV-LCL feeder cells with IL-2, the increased expression of CXCR3 by NK cells led to improved homing and tumor-killing activity against CXCL10-transfected melanoma xenograft mice [151]. Nevertheless, the observations on the CXCR3-mediated accumulation of NK cells in the blood and the average survival rate of MM patients emphasize the need for further studies to determine the therapeutic role of CXCR3 [150].

#### 2.3.4. CXCR1-CXCR2

CXCR1 and CXCR2 are highly expressed by CD56^dim^ NK cells and facilitate their tumor infiltration [78,144]. Their ligands are the cancer cell-expressed chemokines CXCL6 and CXCL8 for CXCR1, and CXCL1–CXCL7 for CXCR2 [51]. In a renal cell carcinoma in vitro study, primary NK cells were retrovirally transduced with CXCR2, resulting in enhanced migration to the tumor sites [152]. Interestingly, the inhibition of CXCR2 in a melanoma mouse study resulted in a reduction in NK cell tumor infiltration and an enhancement of the survival rate of melanoma-bearing mice [153]. In another study, CRISPR/Cas9 gene editing was applied to simultaneously overexpress CXCR2 and IL-2 in NK-92 cells. These results from Gao and colleagues demonstrated enhanced proliferation and chemotaxis to tumor sites, as well as tumor reduction and an increased survival rate in vivo [154]. Moreover, the importance of CXCR1 was observed in ovarian cancer xenograft models when its overexpression via mRNA electroporation enhanced the migration, infiltration, and efficacy of NKG2D-engineered CAR–NK cells [155].

Finally, numerous studies have shown the importance of the fractalkine (CX3CL1) receptor CX3CR1 for the recruitment of NK cells as well as other immune cells like monocytes and dendritic cells [56]. Interestingly, NK cell migration was substantially decreased via CX3CR1 antagonism in an esophagogastric adenocarcinomas study [156].

### 2.4. Cytokine Manipulation

The administration of exogenous cytokines is a standard method for expanding infused NK cells in vivo [6]. To moderate cytokine toxicity and the necessity of multiple cell infusions, researchers have directed their attention towards the genetic modulation of cytokine expression in NK cells [6]. To this extent, Nagashima and colleagues retrovirally transduced NK-92 cells to express IL-2, expanding their in vitro persistence for up to 5 months without the need for additional cytokine administration [31,157]. Primary NK and NK-92 cells have been transduced with retroviral vectors for the expression of IL-2 or IL-15, enhancing NK cell expansion and perseverance in vivo in mice [158]. Researchers have also retrovirally transduced CB-NK cells to express IL-15 together with CAR–CD19, then administered them to a xenograft Raji lymphoma murine model. Based on their results, CB-NK cells restored their function, and overall survival was prolonged [78,159]. The IL-15 transgene has been incorporated into a CAR construct by Daher and colleagues, increasing NK cell proliferation and sustainment in vivo [6]. However, it has been shown that IL-15 is negatively regulated by the cytokine-inducible SH2-containing protein, also known as CIS regulatory element, which is encoded by the CISH gene [17,23]. Blocking CIS activity leads to a significant reduction in the NK cell activation threshold [17]. The importance of the CISH gene knockout for the NK cell metabolic activity in murine in vivo studies against several cancer types, such as prostate, melanoma, and breast cancer, has been demonstrated by Delconte and colleagues [18,160,161]. The absence of CISH regulation of IL-15 resulted in increased levels of IL-15, overcoming this inhibitory obstacle against NK cell perseverance and cytotoxicity [25,160]. CRISPR/Cas9 technology was also utilized by the Kaufman group for the CISH gene deletion in iPSC-derived cells before their differentiation into NK cells, providing NK cells with elevated killing potential and survival in AML in vitro and in vivo studies [23,25,162].

**Table 2 pharmaceutics-16-01143-t002:** Preclinical studies focused on NK cell receptors and their modification for cancer immunotherapy.

NK Cell Receptor	Type of Engineering/Antibody	Method of Engineering	Target	NK Cell Source	Key Outcomes	Year	Ref.
Activating Receptors							
CD16	Overexpression of hnCD16 and IL-2	DNA electroporation	Head and neck, breast, lung, colorectal, and pancreatic cancer	NK-92	In vitro lysis of many tumor cell lines, including lung, breast, head and neck, and colon	2016	[62]
CD16	Knockout of ADAM17 and PD-1	CRISPR/Cas9 electroporation	B-cell lymphoma (Raji cell line)	PB-NK	In vitro enhanced ADCC with rituximab, and elevated IFN-γ levels	2020	[63]
CD16	Overexpression of hnCD16	Lentiviral vectors	B-cell lymphoma, ovarian cancer	iPSC-NK	In vivo enhancement of ADCC with anti-CD20 and anti-HER2 mAbs	2020	[60]
CD16	Overexpression of hnCD16	Retroviral vectors	B-cell leukemia and lymphoma (cell lines and primary patient material)	UCB-NK or PB-NK	Successful expansion and sorting of CD16^+^ cells and in vitro enhancement of ADCC with rituximab or elotuzumab	2024	[64]
CD16 CD38	Overexpression of CD16 and knockout of CD38	CRISPR/Cas9 and mRNA Electroporation	Myeloma (MM.1S, NCI-H929, and EJM cell lines)	iPSC-NK	In vitro and in vivo enhanced ADCC with daratumumab reaching high lysis levels of NCI-H929 myeloma cells and reduction of tumor burden in MM.1S xenograft mouse	2022	[65]
CD16	Overexpression of CD16 in combination with CAR construct	Lentiviral vectors	Triple-negative breast cancer	NK92MI	In vitro enhancement of ADCC with L-ICON immunoconjugate agent and in vivo reduction in tumor growth in cell line- and patient-derived xenograft mouse models	2020	[163]
CD16	Bispecific single domain antibody (VHH) to engage CD16 and EGFR	-	Colorectal cancer (cell lines)	PB-NK	In vitro and ex vivo enhanced activation of NK cells and lysis of EGFR-expressing tumor cell lines	2021	[57]
CD16	BiKE for CD16 and CD33 engagement	-	Leukemia and B-cell lymphoma (HL60 and Raji cell lines)	PB-NK	In vitro activation of NK cells with CD16xCD33 BiKE and ADAM17 inhibition against refractory CD33^+^ AML cells	2013	[70]
CD16	161533 TriKE for CD16 and CD33 engagement and IL-15 production	-	Myelodysplastic syndrome (MDS)	PB-NK	In vitro enhancement of NK cell proliferation and targeting of primary MDS blasts	2018	[71]
CD16	AFM13 bispecific antibody for CD16 and CD30 engagement	-	T-cell lymphoma (Karpas 299 and HuT-78 cell lines and Karpas-NSG mice)	PB-NK or CB-NK	In vitro and in vivo enhancement of NK cell cytotoxicity and cytokine production against CD30^+^ lymphoma targets	2021	[74]
NKG2D	Overexpression of NKG2D and IL-21	DNA delivery via chitosan-based nanoparticles	CT-26-induced solid tumors in Balb/c mice	-	In vivo transfection of tumor cells increased stimulation and migration of NK cells to tumor sites, slowed tumor growth and prolonged the life spam of tumor-expressing mice	2017	[80]
NKG2D	Overexpression of miR-486-5p	Lipofection (HiPerfect Transfection Reagent)	Hepatocellular carcinoma (Huh7 cell line)	PB-NK	In vitro activation of NK cells, induction of NKG2D levels, and enhanced lysis of Huh7 cells	2016	[82]
NKG2D	Overexpression of NKG2D	Lentiviral vectors	Human sarcoma explants and tumor cell lines	NK-92	In vitro activation of NK-92 cells and enhanced degranulation towards sarcoma explants and tumor cell lines	2020	[83]
NKG2D	Overexpression of NKG2D in combination with CAR construct	Retroviral vectors	Myeloid-derived suppressor cells (MDSCs)	PB-NK	In vitro and in vivo increased cytotoxicity against MDSCs, pro-inflammatory cytokine and chemokine production, and tumor infiltration. Exogenous NKG2D was not susceptible to TME compared to endogenous NKG2D	2021	[164]
NKG2D	BiKE for NKG2D and CS1 engagement	-	MM	PB-NK	In vivo activation of NK cells and improved tumor clearance when tested in a xenograft NOD-SCID (NSG) mouse model	2018	[86]
NKG2D	BiKE for NKG2D and HER2 engagement	-	Breast ductal carcinoma (BT-474 cell line)	Human NK (source not specified)	In vitro enhancement of unstimulated NK-cell-mediated killing of BT-474 cells but did not promote the secretion of pro-inflammatory cytokines	2020	[88]
NKG2C	Overexpression of NKG2C and TriKE for NKG2C and CD33 engagement and IL-15 production	Feeder cells	AML (cell lines and blasts)	iPSC-NK	In vitro enhancement of NK cell proliferation, degranulation, and IFN-γ production against AML cell lines and primary AML blasts	2021	[94]
NKp46	FLEX-NK engager for NKp46, CD38, and Glypican-3 (GPC3) engagement	-	Hepatocellular carcinoma (cell lines and spheroids)	PB-NK	In vitro enhancement of antibody-dependent cellular phagocytosis and complement-dependent cytotoxicity of NK cells towards GPC3-expressing tumors	2023	[96]
NKp44	mAb 14-25-9 for NKp44 and PCNA binding inhibition	-	MM	NK92-44-1 (transduced NK-92 cells to express NKp44)	In vitro enhancement of degranulation and IFN-γ production of NK92-44-1 cells against MM primary cells	2022	[98]
NKp30	B7-H6 engagers for NKp30 and EGFR engagement	-	Epidermoid carcinoma and non-small cell lung carcinoma (A431 and A549 cell lines)	PB-NK	In vitro enhancement of IFN-γ and TNF-α production, and cell-mediated cytotoxicity of NK cells against EGFR-expressing tumor cell lines	2021	[44]
NKp30	Engagers for NKp30 and EGFR engagement	-	Epidermoid carcinoma and non-small cell lung carcinoma, colorectal and colon cancer (A431, A549, SW-480, HCT116, and H2030 cell lines)	MNC-NK	In vitro enhancement of NK cell activation and cell-mediated killing of NK cells against EGFR-expressing cell lines	2022	[99]
Inhibitory receptors							
KIR	IPH2101 mAb to block KIR and HLA-I engagement	-	MM (U266 and K562 cell lines)	PB-NK	In vitro enhancement of NK cell survival and activity against AML cell lines	2011	[104]
KIR	Overexpression of KIR2DL1 in combination with a CAR construct (iCARs)	Retroviral vectors	B-cell lymphoma (Raji cell line)	CB-NK	In vitro and in vivo improvement of antitumor-activity when iCARs were administrated in Raji tumor-bearing NSG mice. iCARs prevent trogocytosis-induced self-recognition and fratricide, maintaining tumor recognition and cytotoxicity	2022	[165]
NKG2A	shRNA knockout of NKG2A	Lentiviral vectors	B lymphoblastoid cells	PB-NK	In vitro induction of target cell lysis by 40% compared to NKG2A-expressing NK cells	2008	[110]
NKG2A	Blockade of NKG2A	Protein expression blockers with retroviral vectors	HLA-E tumors	PB-NK	In vivo enhancement of NK cell cytotoxicity against HLA-E tumor-expressing immunodeficient mice	2019	[111]
NKG2A	Knockout of NKG2A	CRISPR/Cas9 nucleofection	MM (primary cells)	PB-NK	In vitro enhancement of cytolytic activity of NKG2A–KO NK cells with no significant difference in cytokine production comparing with NKG2A-expressing NK cells	2022	[112]
NKG2A	Knock out of NKG2A	CRISPR/Cas9 Lentiviral vectors	Metastatic breast cancer	PB-NK	In vivo delay of tumor progression and enhancement of survival in an HLA-E^+^ metastatic breast cancer xenogeneic mouse model	2023	[113]
NKG2A	Monalizumab for blockade of NKG2A and CD94 engagement	-	Chronic lymphocytic leukemia (K562 cell line)	PB-NK	In vitro restoration of NK cell activity against HLA-E-expressing targets, without impacting ADCC	2016	[115]
NKG2A	Knockout of NKG2A in combination with CAR construct	CRISPR/Cas9 Lentiviral vectors	AML	PB-NK	In vivo complete elimination of AML and AML-initiating cells in an AML-xenografted mouse model	2022	[166]
TIGIT	13G6 mAb for TIGIT blockade	-	Colon cancer	-	In vivo restoration of NK cell activity and cytokine production, and prolonged CT26 tumor-bearing mice survival	2018	[116]
TIGIT	Anti-mouse TIGIT (4B1) and aCD266 mAb for the blockade of TIGIT and CD266	-	Melanoma	-	In vitro and in vivo enhancement of NK-cell mediated cytotoxicity and decrease in tumor metastasis in mouse melanoma models	2020	[123]
TIGIT	Knockout of TIGIT	CRISPR/Cas9 electroporation	Pediatric and lung cancer (cell lines and spheroids)	PB-NK	In vitro enhancement of NK-cell cytotoxicity against all tumor cell lines and spheroids tested except CHLA90 that expresses less DNAM-1 and H1975 that is generally susceptible to NK-cell-mediated killing	2023	[125]
TIGIT	CD155 and CD73 targeting	SynNorch Lentiviral vectors	Glioblastoma	iPSC-NK	In vivo decrease in tumor growth by 40% with iNK cells, enhancement of NKp46 and granzyme B in a xenograft glioblastoma mouse model and isolated brain samples	2024	[126]
CD96	anti-CD96 mAb for CD96 blockade	-	Lung cancer	-	In vivo enhancement of survival of mice bearing B16F10 or RM-1 lung metastases	2016	[127]
CD112R	anti-CD112R mAb for CD112R blockade	-	Colon adenocarcinoma	-	In vivo restoration of NK cell and T-cell function, and prolongation of survival of MC38 tumor-bearing mice	2021	[131]
CTLA-4	Ipilimumab for CTLA-4 blockade	-	Melanoma	PB-NK	In vivo inhibition of tumor after treatment of chimeric murine xenograft model with allogeneic NK cells and Ipilimumab	2013	[133]
PD-1	Anti-PD-1 and anti-PD-L1 for blockage of PD-1 and PD-L1	-	HLA-negative cancers	-	In vivo enhancement of NK cell response towards HLA-negative tumor-bearing mice	2018	[136]
TIM-3	Knockout of TIM-3	CRISPR/Cas9 electroporation	Glioblastoma (cell lines)	PB-NK	In vitro enhancement of NK-cell-mediated growth inhibition of GBM cells	2021	[142]
Chemotactic receptors							
CXCR4	Overexpression of CXCR4	mRNA electroporation	-	PB-NK	In vitro enhanced chemotaxis towards SDF-1a, and in vivo increased bone marrow homing in NSG mice	2019	[53]
CXCR4	Knockout of CXCR4 in myeloid cells	Knockout mice	Melanoma	-	In vivo reduction in tumor growth and of FasL-expressing myeloid cells, and enhancement of Fas-expressing NK cells	2018	[54]
CXCR4	Overexpression of CXCR4 in combination with CAR construct	Lentiviral vectors	Leukemia (NALM-6 cell line)	PB-NK	In vitro enhancement of migration towards SDF-1a, and CD16^+^ tumor eradication while retaining functional activity	2020	[146]
CCR7	Overexpression of CCR7	Feeder cells trogocytosis	-	PB-NK	In vitro migration of NK cells towards CCL19 and CCL21 and in vivo enhancement of lymph node homing in athymic nude mice	2012	[148]
CCR7	Overexpression of CCR7and CD16	mRNA electroporation	Chronic myelogenous leukemia and melanoma (K562 and MM.1S cell lines)	PB-NK	In vitro promotion of migration towards CCL19 and enhancement of CD16-mediated ADCC with rituximab	2016	[149]
CCR7	Overexpression of CCR7 and CXCR4	Lentiviral vectors	Colon cancer	NK-92	In vitro and in vivo promotion of migration to colon cells and increased survival of HT-29 tumor-bearing SCID mice	2020	[55]
CXCR3	Overexpression of CXCR3	Feeder cells	Renal cell carcinoma (cell lines) and melanoma (526MEL tumor-expressing mice)	PB-NK	In vitro and in vivo NK cell infiltration to CXCL10-expressing solid tumors, reduction in tumors, and increased survival of CXCL10-positive melanoma xenograft mice	2014	[151]
CXCR2	Overexpression of CXCR2	Retroviral vectors	Renal cell carcinoma (cell lines)	PB-NK	In vitro promotion of NK cell migration to tumor sites with no significant difference in degranulation against K562 cells compared to CXCR2^-^ NK cells	2017	[152]
CXCR2	Overexpression of CXCR2 and IL-2	CRISPR/Cas9	Colon cancer	NK-92	In vitro enhancement of NK cell infiltration to tumor sites and in vivo prolongation of survival and reduction in colon tumor growth in tumor-bearing mice	2021	[154]
CXCR1	Overexpression of CXCR1 in combination with CAR construct	mRNA electroporation	Peritoneal ovarian cancer	PB-NK	In vitro enhancement of NK-cell migration and in vivo infiltration to tumor sites and tumor shrinking in a intraperitoneal xenograft NSG mouse model	2020	[155]

**Table 3 pharmaceutics-16-01143-t003:** Recent clinical studies focused on NK cell receptors and their modification for cancer immunotherapy.

NK Cell Receptor	Product/Study	Malignancy	NK Cell Source	Sponsor	Location	Status	Clinical Phase	Year	Key Outcomes	ClinicalTrials.gov Identifier
Activating Receptors										
CD16	AFM13	Hodgkin Lymphoma	Intravenous infusion	Affimed GmbH	Houston TX USA, Würzburg GER	Completed	I	2010	In total, 3 of 26 patients achieved partial remission (11.5%) and 13 patients achieved stable disease (50%), with an overall disease control rate of 61.5%	NCT01221571
CD16	AFM24, SNK01	Squamous Cell Carcinoma of Head and Neck, Non-Small-Cell Lung Carcinoma, Colorectal Neoplasms, Advanced Solid Tumor, Refractory Tumor, and Metastatic Tumor	Autologous SNK01	NKGen Biotech, Inc.	Los Angeles CA, Chicago IL USA	Terminated	I/II	2021	Patients had a manageable safety profile, and SNK01 monotherapy has also shown to be well-tolerated in patients with rapidly progressive solid tumors	NCT05099549
CD16	ha-NK, Avelumab, N-803	Merkel Cell Carcinoma	NK-92	ImmunityBio, Inc.	San Francisco CA, Miami FL, Saint Louis MO USA	Terminated	II	2020	Did not meet recruitment goal	NCT03853317
CD16	haNK, Avelumab, Bevacizumab	Pancreatic Cancer	NK-92	ImmunityBio, Inc.	El Segundo CA USA	Active, not recruiting	I/II	2017	-	NCT03329248
						Terminated		2017	The study was terminated early due to low enrollment. Safety data showed a 2/4 (50%) all-cause mortality.	NCT03387098
						Active, not recruiting		2018	-	NCT03586869
CD16	haNK, Avelumab, Bevacizumab	Squamous Cell Carcinoma	NK-92	ImmunityBio, Inc.	El Segundo CA USA	Terminated	I/II	2018	The study was terminated early due to low enrollment. Safety data showed a 4/4 (50%) all-cause mortality.	NCT03387111
CD16	FT516 (hnCD16), CD20 Ab/PD-L1 Ab	B-cell Lymphoma and Acute Myeloid Leukemia/Solid Tumors	iPSCs	Fate Therapeutics	Phoenix AZ, San Diego CA, Minneapolis MN USA	Terminated	I	2019	The study was terminated by the Sponsor	NCT04023071
								2020		NCT04551885
CD16	FT516 (hnCD16), Enoblituzumab, IL-2	Ovarian, Fallopian Tube, and Primary Peritoneal Cancer	iPSCs	Masonic Cancer Center, University of Minnesota	Minneapolis MN USA	Completed	I	2021	-	NCT04630769
CD16, CD38	FT538 (hnCD16, CD38KO), IL15RF, Daratumumab	Multiple Myeloma and Acute Myeloid Leukemia/Solid Tumors	iPSCs	Fate Therapeutics	Denver CO, Minneapolis MN, Saint Louis MO, Hackensack NJ, Nashville TN USA	Terminated	I	2020	Interim Outcomes: Administration of FT538 cells in combination with daratumumab was safe and well tolerated without indication of CRS, neurotoxicity, or GvHD	NCT04614636
								2021	This study was terminated by the Sponsor	NCT05069935
CD16, CD38	FT538 (hnCD16, CD38KO), Daratumumab, Fludarabine, Cyclophosphamide	Acute Myeloid Leukemia	iPSCs	Masonic Cancer Center, University of Minnesota	Minneapolis MN USA	Active, not recruiting	I	2021	FT538 combined with daratumumab has been well-tolerated in a heavily pre-treated patient group, showing expected toxicities and some signs of efficacy	NCT04714372
NKG2D	NAKIP-AML, Talazoparib	Acute Myeloid Leukemia	Haploidentical human allogeneic NK cells	German Cancer Research Center	-	Not yet recruiting	I/II	2024	-	NCT05319249
NKG2C and PD-1	Dasatinib	Chronic Myeloid Leukemia	CMV-activated NKG2C^+^NK	Nanfang Hospital of Southern Medical University	Guangzhou CHI	Unknown	Obs.	2021	-	NCT04991532
NY-ESO-1 TCR/IL-15	NY-ESO-1 TCR/IL-15 NK cells	Multiple Myeloma	CB-NK	M.D. Anderson Cancer Center	Houston TX USA	Recruiting	I/II	2023	-	NCT06066359
KIR	IPH2101	Multiple Myeloma, Myeloma, and Smoldering Multiple Myeloma	Intravenous infusion	National Cancer Institute (NCI)	Bethesda MD USA	Terminated	II	2010	Lack of patients meeting the defined primary objectives	NCT01248455
Inhibitory Receptors										
CTLA4	Ipilimumab, Cetuximab, CIML NK cells, N-803	Squamous Cell Carcinoma of the Head and Neck, Recurrent Head and Neck Squamous Cell Carcinoma	CIML NK	Dana-Farber Cancer Institute	Boston MA USA	Recruiting	I	2020	Initial Outcomes: Allogeneic CIML NK cells combined with N-803 may promote tumor regression in patients with advanced head-and-neck cancer	NCT04290546
PD-1	SMT-NK Pembrolizumab	Biliary Tract Cancer	Allogeneic SMT-NK	SMT bio Co., Ltd.	Seoul KOR	Completed	I/II	2019	In phase 2a, 126 adverse events (AEs) were observed in 29 patients (85.3%). Severe AEs occurred in 16 patients (47.1%), but no dose-limiting toxicities were reported. The overall response rate (ORR) was 17.4% in the full-analysis set and 50.0% in the per-protocol set	NCT03937895
PD-1	NK cells Sintilimab	Non-small Cell Lung Cancer	Autologous PBMCs	The First Hospital of Jilin University	Changchun CHI	Completed	II	2019	Autologous NK cells combined with sintilimab demonstrated promising antitumor activity and had an acceptable safety profile in advanced NSCLC patients. No unexpected AE were observed	NCT03958097
PD-1	Pembrolizumab, DC-NK cells	Solid Tumors	Intravenous infusion	Allife Medical Science and Technology Co., Ltd.	-	Unknown	I	2019	-	NCT03815084
PD-1	NK and DC cells, Pembrolizumab, Nivolumab, Sintilimab, Toripalimab, Camrelizumab, Tislelizumab	Digestive Carcinoma, Gastrointestinal Tumors	Autologous NK cells	China Medical University	-	Not yet recruiting	II	2022	-	NCT05461235
PD-1	COH06 Azetolizumab	Several types of Non-Small cell Lung carcinoma	CB-NK	City of Hope Medical Center	Duarte CA USA	Active, not recruiting	I	2022	-	NCT05334329
PD-1	D-CIK cells, Axitinib	Renal Metastatic Cancer	PBMCs	Sun Yat-sen University	Guangzhou CHI	Unknown	II	2018	-	NCT03736330
PD-1	CCICC-002b, CIK cells, Sintilimab	Non-small cell lung cancer	Autologous CIK cells	Tianjin Medical University Cancer Institute and Hospital	Tianjin CHI	Unknown	II	2021	-	NCT04836728
PD-1	D-CIK, anti-PD-1	Refractory Solid Tumors	PBMCs	Sun Yat-sen University	Guangzhou CHI	Unknown	I/II	2016	This study indicated enhanced antitumor immunity following combination treatment, particularly in patients with significant long-term survival benefits. In contrast, those with minimal survival benefit exhibited a higher proportion of peripheral CD8+TIM3+ T cells and a lower serum-level immunostimulatory cytokine profile	NCT02886897
PD-1	D-CIK and Pembrolizumab	Lung cancer neoplasms	Autologous PBMCs	Capital Medical University	Beijing CHI	Completed	I/II	2016	-	NCT03360630
PD-1	Anti-PD-1 P-GEMOX	High-risk Extranodal NK/T-cell lymphoma	Intravenous infusion	Cancer Institute and Hospital, Chinese Academy of Medical Sciences	Beijing CHI	Recruiting	II	2021	-	NCT05254899
PD-1	Pembrolizumab	NK/T cell lymphoma	Intravenous infusion	The University of Hong Kong	Hong Kong HKG	Unknown	II	2016	-	NCT03021057
PD-1	Merck NK-IIT Pembrolizumab	Melanoma	Intravenous infusion	Nina Bhardwaj	New York NY USA	Terminated	II	2017	Did not meet recruitment goal	NCT03241927
PD-1	SHR-1210 CIK cells	Renal Cell Carcinoma	Autologous CIK cells	Tianjin Medical University Cancer Institute and Hospital	Tianjin CHI	Unknown	II	2019	-	NCT03987698
PD-1	Toripalimab	Extranodal NK/T-cell lymphoma	Intravenous infusion	Beijing Tongren Hospital	-	Unknown	II	2020	-	NCT04338282
PD-1	Anti-PD-1 Chidamide Lenalidomide Etoposide	Relapsed or refractory NK/T-cell lymphoma	Intravenous infusion	Mingzhi Zhang	Zhengzhou CHI	Unknown	IV	2019	The 12-month progression-free survival (PFS) rate was 86.8%. All 19 patients experienced treatment-related adverse events (TRAEs), with 4 patients (21.1%) reporting immune-related AEs, including grade 1 hypothyroidism	NCT04038411
PD-1	Anti-PD-1 Pegaspargase	Extranodal NK/T-cell lymphoma	Intravenous infusion	Ruijin Hospital	Shanghai CHI	Unknown	II	2019	The combination of pegaspargase and sintilimab is effective and safe for treating advanced-stage NKTCL, with potential benefits in targeting fatty acid metabolism and CTLA-4 to overcome treatment resistance	NCT04096690
PD-1	SHR1210 Apatinib	NK/T-cell lymphoma	Intravenous infusion	Peking University	-	Unknown	II	2018	The overall response rate (ORR) was 30%, with 10% of patients achieving a complete response. The median progression-free survival (PFS) was 5.6 months, and the median overall survival was 16.7 months	NCT03701022
PD-1	Toripalimab Chemoradiotherapy	NK/T-cell lymphoma	Intravenous infusion	Sun Yat-sen University	Guangzhou CHI	Recruiting	III	2020	A total of 714 NKTCL patients were included. The median overall survival (OS) was 36 months, and cancer-specific survival (CSS) was 57 months	NCT04365036
PD-1	CAR2BRAIN, NK-92/5.28.z Ezabenlimab	Glioblastoma	NK-92	Johann Wolfgang Goethe University Hospital	Frankfurt, Mannheim, Mainz GER	Active, not recruiting	I	2017	Study Objectives: Assessing for safety and tolerability to establish the maximum tolerated dose	NCT03383978
PD-L1	QUILT-3.060 NANT haNK	Pancreatic Cancer	NK-92	ImmunityBio, Inc.	El Segundo CA USA	Active, not recruiting	I/II	2017	Initial Outcomes: Beneficial to patients with NSCLC	NCT03329248
PD-L1	QUILT-3.064, PD-L1 t-haNK	Advanced or metastatic solid tumors	NK-92	ImmunityBio, Inc.	El Segundo CA USA	Active, not recruiting	I	2019	-	NCT04050709
PD-L1	Sacituzumab, PD-L1 t-haNK, N-803	Advanced Triple Negative Breast Cancer	NK-92	ImmunityBio, Inc.	Newport Beach CA USA	Terminated	I/II	2021	Low enrollment	NCT04927884
PD-L1	PD-L1 t-haNK, N-803, Aldoxorubicin	Pancreatic Cancer	NK-92	ImmunityBio, Inc.	El Segundo CA, Newport beach CA, East Brunswick NJ USA	Active, not recruiting	II	2020	-	NCT04390399
PD-L1	QUILT-3.063 Avelumab haNK	Merkel Cell Carcinoma	NK-92	ImmunityBio, Inc.	San Francisco CA, Miami FL, Saint Louis MO USA	Terminated	II	2020	Did not meet recruitment goal	NCT03853317
PD-1/PD-L1	QUILT-3.055, Anti-PD-1, Anti-PD-L1, PD-L1 t-haNK	Cancers previously treated with PD-1/PD-L1 Immune Checkpoint Inhibitors	NK-92	ImmunityBio, Inc.	Anchorage AK, Hot Springs AR, El Segundo CA USA	Active, not recruiting	II	2018	Initial Outcomes: N803 shows low toxicity and promising efficacy in halting progression and inducing durable stable disease in patients who had previously progressed on various tumor types and CPI regimens	NCT03228667
TGF-β/NR3C1	CB-NK-TGF-betaR2-/NR3C1^-^	Glioblastoma	CB-NK	M.D. Anderson Cancer Center	Houston TX USA	Recruiting	I	2023	-	NCT04991870
Chemotactic Receptors										
CXCR4	Revolution CXCR4 antagonists in combination with Nivolumab	Metastatic Renal Cell Carcinoma	PBMCs	National Cancer Institute, Naples	Naples ITA	Recruiting	I	2016	Baseline NK activity and early detection of CXCR4-dependent reversal of Treg suppressive function at two weeks are significant indicators of response in mRCC patients treated with Nivolumab	NCT03891485

### 2.5. Combining CAR–NK with NK Receptor Engineering

The CAR approach has been established over the past few decades as a breakthrough cell manipulation technology for tumor targeting, with CAR–T constructs already receiving FDA and EMA approval [5,24]. A CAR is a synthetic protein construct, usually derived from the combination of four different domains: an extracellular antigen-binding domain, a hinge region, a transmembrane domain, and an intracellular signaling domain (Figure 4) [167,168,169]. The target-binding specificity is determined by the antigen-binding domain, which in most cases includes a single-chain variable fragment (scFv), deriving from antibodies, and more rarely, a native protein or a peptide [167,170]. CD19 has emerged as the predominant CAR target for hematological malignancies, demonstrating promising results in terms of cell expansion and patient remission rates [22,171,172].

As previously discussed in this review, NK cells present a significant advantage in addressing the toxicity challenges associated with T cell therapies. Similarly, CAR–NK cells have demonstrated the ability to overcome CRS, GvHD, and neurotoxicity, which are commonly linked to CAR–T cells, as CAR–NK express a distinct cytokine profile and, due to MHC-I recognition, do not attack self-tissues [172]. Moreover, while CAR–T cells primarily rely on CAR-mediated recognition of specific antigens, CAR–NK cells target tumors through both CAR-dependent and endogenous receptor-mediated methods, including tumors that lack HLA expression [173]. The absence of HLA-matching requirements enables the development of “off-the-shelf” CAR–NK products, which could reduce the substantial logistical costs associated with CAR–T cell therapies [174]. The diverse tumor recognition mechanisms of CAR–NK cells offer the potential for targeting both hematological malignancies, such as lymphomas, and solid tumors, including colorectal, breast, and pancreatic cancer [175]. 

This is why CAR–NK cells have gained a lot of attention recently as clinical trials for hematological and solid tumors are growing rapidly [176]. CAR–NKs have been developed for solid tumor elimination targeting EGFRvIII, HER2, and mesothelin, as these molecules are present in various cancer types, such as glioblastoma, colorectal, ovarian, and breast cancer [22]. Other types of tumor antigens have been targeted as well with recent CAR–NK applications, such as CD20, CD22, CD33, CD123, and GD2 [169,177]. The hinge region of the CAR–NK construct, usually derived from CD8, CD28, or IgG4, is responsible for exposing the antigen-binding domain on the cell surface [167,178]. The transmembrane domain anchors the CAR construct on the cell membrane and links it to the intracellular signaling domain [167,170].

The intracellular signaling domain stimulates NK cell cytotoxic activity and is the most thoroughly studied domain in the CAR-engineering field [24,167,169]. CARs have evolved through four different generations, based on variations in the design of the intracellular domain [167]. Conventional first-generation CAR–NKs, similarly to CAR–Ts, include only CD3ζ as a signaling domain, while second- and third-generation CAR–NKs include one or two co-stimulatory domains, mostly CD28 and 4-1BB (CD137), derived from CAR–T cells [167,170]. Recently, novel CAR–NKs have been developed with the inclusion of DNAX-activation proteins DAP12 and DAP10, which are responsible for the stimulation of NKp44, activating KIR receptors (KIR2DS and KIR3DS), and NKG2C [24,173,179]. Finally, fourth-generation CARs consist of engineered cells to self-produce cytokines such as IL-2, IL-15, or IL-12, thereby enhancing their proliferation, migration, and activation towards tumor cells [168,170,176,180]. In this case, a transgenic response modulator is integrated into the CAR construct for the simultaneous production of immune modulators such as cytokines, chemokines, and suicide genes, which recruit other components of the immune system, leading to an enhanced antitumor response [176]. Although fourth-generation CARs are less investigated in NK cells compared to T cells, recent studies employing CD19-CAR/IL-15 NK-92 cells have shown increased cytotoxicity in vitro and in a xenograft mouse model [181,182].

NK cell receptor engineering, parallel to CAR technology, represents an alternative approach to augment CAR-mediated cytotoxicity while simultaneously modulating the regulatory mechanisms of activating and inhibitory receptors. NKG2D and PD-1 are the only NK receptors currently involved in CAR–NK clinical trials [178]. NKG2D ligands are expressed by several tumor types, making this activating receptor a promising tool for applications targeting hematological malignancies and solid tumors [168,173,178,183]. Besides inducing CAR–NK cytotoxicity towards antigens expressed by various tumor types, NKG2D–CAR–NKs have been designed by Parihar and colleagues to ameliorate NK cell persistence within the immunosuppressive TME by eradicating MDSCs [164]. In addition, PD-L1 CAR–NK cells have been shown to successfully target and eliminate tumors and are currently being evaluated in a phase II clinical trial for HNSCC and gastroesophageal junction cancer [184]. However, researchers are exploring alternative NK receptors in recent preclinical studies. For instance, Li and colleagues developed KIR-based inhibitory CAR–NK cells to overcome the negative effect of trogocytosis in activating CAR–NK cells [165]. In contrast, another group knocked out the inhibitory receptor NKG2A by means of CRISPR/Cas9 technology to potentially enhance the efficiency of CD33-CAR–NK cells in AML [166]. Moreover, Zhiwei Hu’s research has shown interesting results in targeting triple-negative breast cancer using CAR-engineered NK cells co-expressing CD16 for simultaneous surface antigen targeting and ADCC in combination with L-ICON. The efficacy of CAR–NK cells was significantly enhanced in combination with L-ICON-mediated ADCC both in vitro and in vivo using patient’ tumor-derived xenograft mouse models [163]. All the aforementioned methods to introduce new genetic material into the NK cells for the expression of the fusion CAR–NK receptor have been utilized by various research groups in preclinical studies [22]. Table 4 summarizes the CAR–NK studies in combination with NK cell receptor engineering, which are currently in the stage of clinical trials.

## 3. Conclusions

Although NK cells have emerged as alternative source of effector cells addressing the toxicity challenges linked to T cell therapies and CAR–T cells, significant limitations remain, namely their low proliferation and persistence in the body, their low infiltration and efficacy in solid tumors, the immunosuppressive TME, and off-target toxicities. This review provides an in-depth look at the preclinical and clinical advancements in endogenous NK receptor engineering and blocking, indicating their potential impact on cancer immunotherapy. These studies highlight the versatile anti-tumor capability of NK cells while underscoring the necessity of these modifications to overcome the barriers typically associated with NK cell therapeutics.

Following ACT, the in vivo proliferation and persistence of NK cells are restricted and highly dependent on the selected source. To date, NK-92 remains the only immortalized NK cell line that has been used in clinical settings due to its enhanced proliferation rate. However, NK-92 lack critical activating receptors, such as CD16 which mediates ADCC and enhances cytotoxicity [23]. On the other hand, primary NK cells can be obtained as allogeneic products by several sources, including peripheral blood, cord blood, or iPSCs, and have been widely applied in most clinical studies [22]. Ex vivo expansion of NK cells is challenging and often results in diminished cytotoxicity. Therefore, repeated injections are needed to maximize the response, leading to elevated costs and inconvenience for patients [185]. Although the low persistence of infused cells can serve as a safety mechanism in case of life-threatening side effects, it also restricts the efficacy of such treatments, as infused NK cells are typically detectable for a few weeks after ACT [186]. Independent injections of cytokine analogs are known to enhance NK cell in vivo proliferation; therefore, current research is focusing on ex vivo genetic modification of NK cells to stably produce such cytokines [159]. For example, engineered NK cells to overexpress IL-2 or IL-15 have significantly enhanced in vivo perseverance in murine models [158]. Additionally, the deletion of the CISH gene, coding the CIS regulatory element of IL-15, is an alternative approach of preclinical studies to promote in vivo persistence and cytotoxicity of NK cells [160,162]. Moreover, cytokine expression has been integrated as an engineering strategy for CAR–NK cells, as fourth-generation CARs enclose a transgenic response modulator to stably produce endogenous cytokines. Studies have shown that IL-15 expression in combination with a CD19-CAR construct can support NK cell function in vivo [159,176]. Promising preclinical studies from MD Anderson Cancer Center using CD19–CAR/IL-15 (CAR19/IL-15) from CB-NK cells have led to the initiation of a phase I/II clinical trial targeting CD19^+^ B lymphoid malignancies, reporting overall response rates ranging from 32% to 68% [clinicaltrials.gov ID: NCT03056339]. Notably, CAR19/IL-15 cells derived from a CB unit with nucleated red blood cells and less than 24 h collection-to-cryopreservation time exhibited superior antitumor activity without treatment-related toxicity, underscoring the importance of NK cell expansion and priming conditions [171]. 

NK cell therapeutics have demonstrated significant clinical responses in patients with hematological malignancies [187,188,189]. However, their effectiveness in solid tumors, such as non-small cell lung carcinoma, renal cell carcinoma, and colorectal cancer, remains limited [190]. This is primarily attributed to insufficient tumor infiltration and the immunosuppressive effects of the TME on NK cell cytotoxicity [191]. NK cells express a wide array of chemokine receptors which allow for cell migration and infiltration into tumors. However, the processes of expansion or cryopreservation of primary NK cells often result in the downregulation of these receptors, leading to insufficient chemotaxis and reduced efficacy in targeting solid tumors [191]. Engineering of NK cells to overexpress chemokine receptors, including CXCR3, CXCR4, and CCR7, has been investigated in preclinical studies. Noteworthy examples include the overexpression of CXCR2 in NK cells targeting renal cell carcinoma and the combined overexpression of CXCR4 and CCR7 for colon cancer. Both approaches demonstrated enhanced migration to tumor sites and prolonged survival of tumor-bearing mice [55,152]. The genetic modification of NK cell to overexpress chemotactic receptors has not yet been explored in clinical settings. However, the promising outcomes observed in the preclinical studies discussed in this review offer valuable insights that can benefit the design of future clinical trials. 

The immunosuppressive and hypoxic TME is another bottleneck of NK cell therapies in both hematological malignancies and solid tumors. The TME exerts a dual inhibitory effect on NK cell function by hindering activating signals and concurrently upregulating the expression of inhibitory receptors and regulatory molecules [185]. To overcome the inhibitory influence of the TME and restore NK cell functionality, current research focus on engineering strategies aims at enhancing the expression of key activating receptors and downregulating inhibitory receptors that impair NK cell activation and cytotoxicity. Among the activating receptors, CD16 and NKG2D appear to be the most preclinically and clinically studied. ADCC is undoubtedly one of the most crucial activating mechanisms of NK cells, and the discovery of therapeutic mAbs has significantly advanced the field. The upregulation of CD16 has already confirmed clinical efficacy in combination therapies with tumor-targeting mAbs. While CD16-engineered NK cells have been well-tolerated in patients, mortality is also observed in some clinical trials from ImmunityBio, Inc, potentially deriving from the combination with tumor-targeting antibodies. Moreover, overexpression of NKG2D, which is also included in many CAR–NK constructs, exhibit promising preclinical results. Recently designed NKG2D–CAR–NKs have been shown to enhance NK cell persistence within the immunosuppressive TME of MDSCs [164].

The blockade of inhibitory receptors has also been shown to enhance NK cell cytotoxicity counteracting the suppressive effects of the TME. Specifically, KIR and PD-1 inhibitory receptors have undergone extensive research and have been targeted with both ICIs (lirilumab and nivolumab, respectively) and knockout techniques. Several other inhibitory receptors were targeted with specific mAbs, such as monalizumab for NKG2A/CD94 or anti-TIM-3 antibodies, as monotherapy or in combination therapies, although their clinical efficacy could be re-evaluated. Other receptors, such as CTLA-4, TIM-3, TIGIT, and CD96, should be studied more thoroughly for their potential clinical targeting for cancer immunotherapies. Moreover, combination therapies have already been applied in preclinical studies, but their clinical translation has mostly focused on their effect on T cells. Finally, the novel breakthrough of CRISPR/Cas9 technology has been successfully employed in several preclinical studies but its application in clinical settings remains limited. Nonetheless, this technology holds significant potential for future clinical studies, particularly for the ex vivo and in vivo genetic manipulation of NK cells.

Overall, the preclinical studies discussed in this review have highlighted the potential of NK cell engineering in addressing the current limitations of NK cell therapeutics in cancer immunotherapy. However, as most clinical trials employing these methods are still in early phases, a comprehensive understanding of the benefits of receptor engineering has yet to be fully established. Future advancement in this field should prioritize optimizing the conditions for isolating, expanding, culturing, and cryopreserving NK cells to preserve their functionalities and to prolong their persistence on clinical applications. The careful selection of donors and phenotypes for “off-the-shelf” NK cell products could contribute to prolonged NK cell persistence in patients [192]. Moreover, harnessing NK chemotactic receptors is pivotal for improving NK cell migration and tumor infiltration, especially in the context of solid tumors. However, this approach has only been explored preclinically in vitro and in vivo. Investigating the potential of NK cells with enhanced anti-tumor chemotaxis represents a promising approach for future clinical studies. Additionally, the co-expression of chemotactic NK cell receptors alongside four-generation CAR constructs presents an intriguing direction, potentially increasing both the cytotoxicity and chemotaxis of NK cells against solid tumors.

As the field of CAR–NK cells continues to advance rapidly, further exploration of NK cell receptors, either as extracellular binding domains or as intracellular signaling domains, offers a promising alternative to strategies derived from CAR–T. To date, only NKG2D and PD-1 have been clinically investigated in combination with CAR constructs. Nevertheless, preclinical studies that combine CAR–NK cells with modified NK cell receptors, such as CD16, NKG2A, and KIR, pave the way for future clinical development [163,165,166]. Additionally, the already applied concept of dual CAR–Ts, which targets two different tumor-expressed antigens simultaneously, resulting in increased resistance to immunosuppression and overcoming tumor immune escape, should also be considered for CAR–NKs [24]. 

Although NK cell therapy is generally considered safer than T cell therapies, off-target toxicities have been observed in some clinical trials involving infused NK cells. These toxicities are commonly reported in studies where NK cell transfers are combined with other therapeutic agents, such as mAb, ICIs, engagers, or other drugs. Consequently, it remains unclear whether the adverse events derive from NK cell transfers or from the combination with other therapeutic agents. However, in a recent clinical study on advanced non-small cell lung carcinoma, one patient developed CRS after receiving three cycles of CAR–NK92 cells [clinicaltrials.gov ID: NCT03656705], with a reported 10-fold increase in IFN-γ and TNF-α levels. The clinicians attributed the CRS to a potential overactivation of NK cells or to the recognition of CAR–NK92 cells as foreign substances [193]. Future evaluation of the reasons behind the occurrence of NK-cell-mediated toxicities is crucial for the development of safe NK-cell-NK-cell-based therapies.

In addition to safety and risk evaluation, further ethical considerations are essential to the development and application of NK cell therapies. Similar to T cells therapeutics and CAR–T cells, NK cell products are often costly and not widely accessible to all populations. Although addressing the limitations of safety and efficacy of effector cells is important, it is also crucial for scientists to ensure that these therapies can and will be available globally. Moreover, ethical concerns arise from the genetic engineering of NK cells, particularly when using transduction or CRISPR/Cas9 technologies, which often result in irreversible modifications. The exploration of transient engineering methods could hold an ethical alternative for modified NK cell products. 

Regarding the methods of engineering, viral DNA transduction and RNA electroporation are the most well-established delivery methods for NK cell receptor modification that have reached clinical development. However, to avoid the toxicity and potential increased cancer risk of long-persistent engineered cell therapies, there has been a recent shift in the field’s interest towards the promising use of LNPs for the transfection of immune cells. Although there are currently no NK cell receptor-related clinical trials reporting the use of LNPs, there is an ongoing trend towards their application through several preclinical studies [35,36,194]. It is anticipated that this delivery method will be widely applied in the field of cancer immunotherapy in the near future [195,196,197,198]. 

Overall, the engineering and modification of NK cell receptors represents a powerful and promising strategy for addressing the current limitations encountered in NK-cell-NK-cell-based immunotherapies targeting both solid and hematological malignancies. Continued clinical evaluation of these approaches is anticipated, offering the potential for significant advancements in the field of cancer immunotherapy. 

## Figures and Tables

**Figure 1 pharmaceutics-16-01143-f001:**
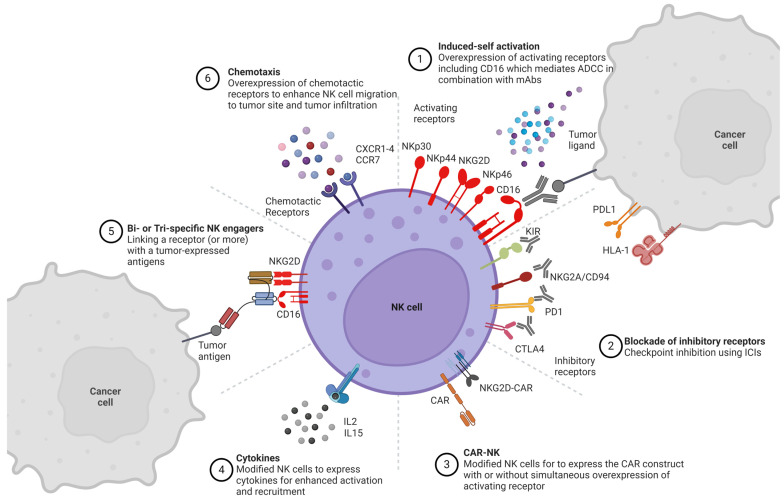
Schematic representation of the key NK-cell-NK-cell-based strategies for cancer immunotherapy. (1) Overexpression of surface-activating receptors for enhanced cell activation and cancer antigen recognition. CD16 overexpression to boost the antibody-dependent cellular cytotoxicity (ADCC) function in combination with therapeutic monoclonal antibodies (mAbs). (2) Blockade of inhibitory receptor interaction with their ligands to ignite NK cell cytotoxicity using immune checkpoint inhibitors (ICIs). (3) Adoptive cell transfer (ACT) of modified NK cells with chimeric antigen receptor (CAR) constructs with or without the simultaneous overexpression of receptor repertoire. (4) Cytokine administration and modified NK cells to endogenously express cytokines for enhanced stimulation. (5) Use of bi- and tri-specific killer engagers for the simultaneous targeting/linking of NK cell receptors and tumor-associated antigens. (6) NK cell chemotactic receptor activation for tumor site migration and subsequent infiltration. Created with BioRender.

**Figure 2 pharmaceutics-16-01143-f002:**
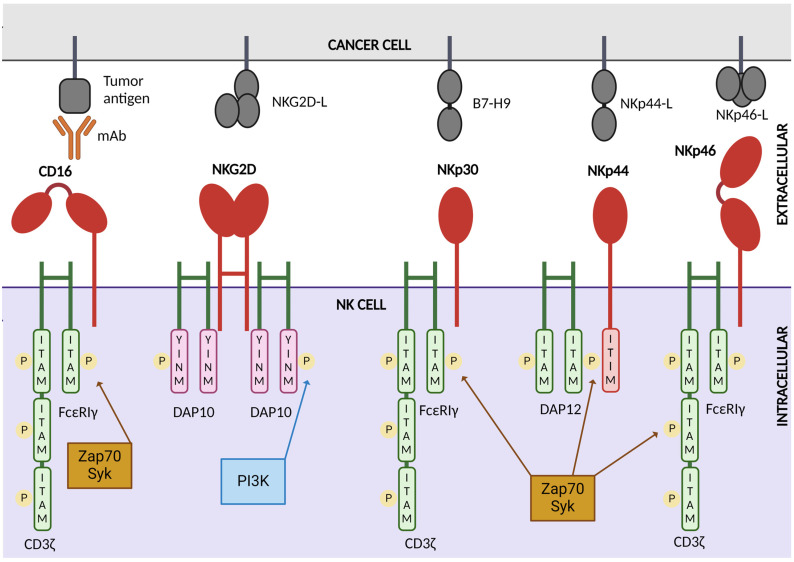
Schematic representation of key activating receptors of natural killer cells, their intracellular domains, and their respective tumor antigens. CD16 binds to the Fc region of IgG monoclonal antibodies that are bound to tumor antigens, initiating the antibody-dependent cellular cytotoxicity function of NK cells. The receptors CD16, NKp30, and NKp46 signal through the phosphorylation of immunoreceptor tyrosine-based motifs (ITAMs) within their associated intracellular domains, CD3ζ and FcεRIγ, mediated by Zap70 and Syk kinases. Similarly, NKp44 transmits signals through the phosphorylation of ITAMs of DAP12 by Zap70 and Syk kinases. The receptor NKG2D signals via two DAP10 intracellular domains, which employ tyrosine–isoleucine–asparagine–methionine (YINM) that is phosphorylated by phosphoinositide 3-kinase (PI3K). Created with Biorender.

**Figure 3 pharmaceutics-16-01143-f003:**
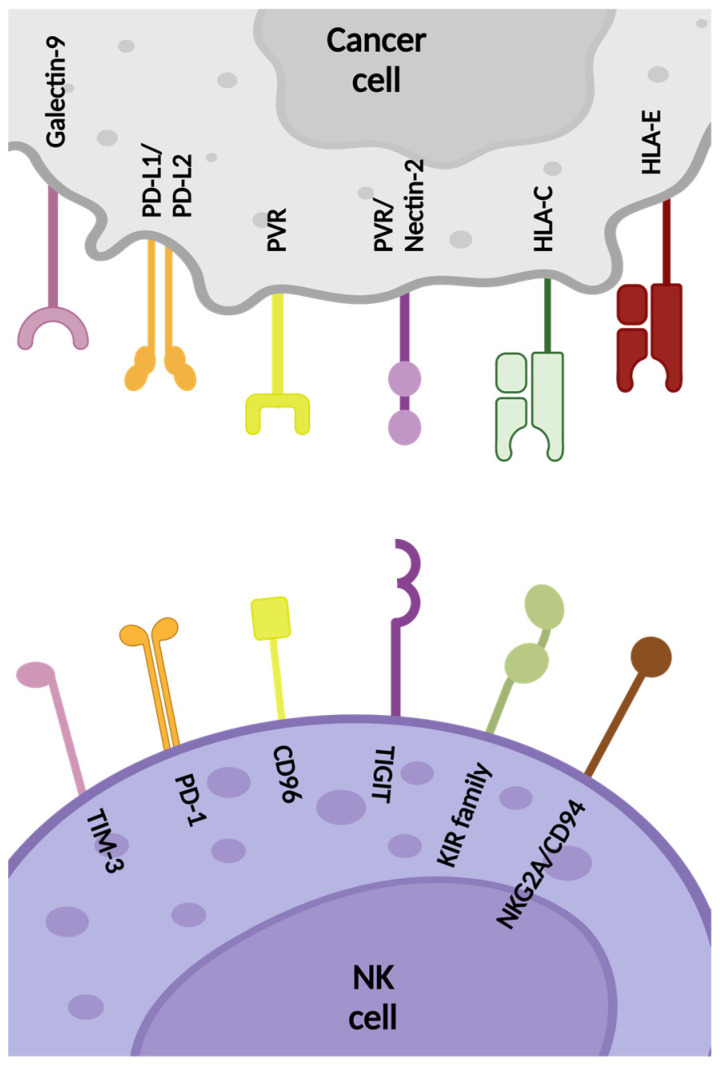
Schematic representation of key inhibitory receptors expressed on the surface of natural killer (NK) cells with their corresponding tumor antigens. Created with Biorender.

**Figure 4 pharmaceutics-16-01143-f004:**
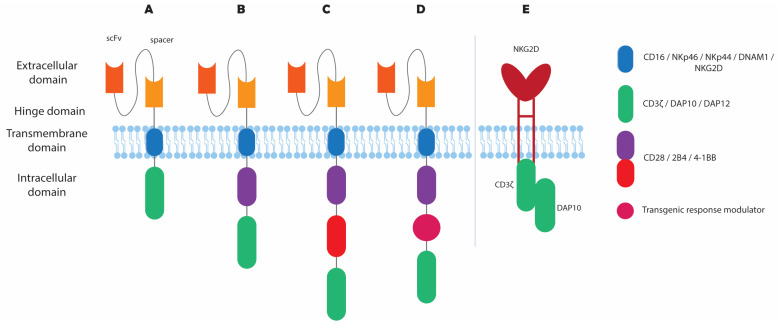
Illustration of the different generations of chimeric antigen receptor (CAR)-NK cells. (A) First-generation CAR–NKs containing an extracellular antigen-binding domain, a hinge domain, a transmembrane domain of either CD16, NKp46, NKp44, DNAM1, or NKG2D, and an intracellular signaling domain of either CD3ζ, DAP10, or DAP12, (B) Second-generation CAR–NKs with the addition of co-stimulatory domains of either CD28, 2B4, or 4-1BB, (C) Third-generation CAR–NKs with the addition of a second intracellular stimulatory domain of either CD28, 2B4, or 4-1BB, (D) Fourth-generation CAR–NKs with the inclusion of a transgenic response modulator for the expression of cytokines, and (E) NKG2D CAR–NK cells, with NKG2D expressed in the extracellular domain, and CD3ζ and DAP10 expressed intracellularly [34,169]. Created with Biorender.

**Table 1 pharmaceutics-16-01143-t001:** List of the most investigated NK cell receptors for cancer immunotherapy and their ligands.

Receptor/Molecule	Ligand/Mode of Action	References
**Activating Receptors**		
CD16 (FcγRIII)	IgG-ADCC	[25]
NKG2D	MHC-I, MICA, MICB, ULBPs	[16,37]
NKG2C	HLA-E	[38]
NKp46 (NCR1, CD335)	Viral hemagglutinins	[39]
NKp44 (NCR2, CD336)	Viral hemagglutinins, Nidogen-1, PCNA, 21spe MML5	[39,40,41,42,43]
NKp30 (NCR3, CD337)	B7–H6, BAT3, pp65	[39,44]
**Inhibiting Receptors**		
KIR family	HLA-A,B,C	[11,24]
NKG2A/CD94	HLA-E	[45]
TIGIT	PVR (CD155), Nectin-2 (CD112)	[13,39,46]
TIM3	Galectin-9	[27,47,48]
PD-1	PD–L1, PD–L2	[28,39]
CTLA-4	B7-1 (CD80), B7-2 (CD86)	[49]
CD96	PVR (CD155)	[39,50]
**Chemotactic Receptors**		
CXCR1	CXCL6, CXCL8	[51]
CXCR2	CXCL1–CXCL7	[6,51]
CXCR3	CXCL9, CXCL10, CXCL11	[23,52]
CXCR4	CXCL12 (or SDF-1a)	[53,54]
CCR7	CCL19, CCL21	[55]
CX3CR1	CX3CL1	[56]

**Table 4 pharmaceutics-16-01143-t004:** Recent CAR–NK clinical trials including NK cell receptors in the CAR construct.

NK Cell Receptor	Product Name	Malignancy	NK Cell Source	Sponsor	Location	Status	Clinical Phase	Year	Key Outcomes	ClinicalTrials.gov Identifier
NKG2DL	NKG2D-CAR–NK92 cells	Relapsed/Refractory Solid Tumors	NK-92	Xinxiang medical university	Xinxiang CHI	Recruiting	I	2023	-	NCT05528341
NKG2DL	NKG2D CAR–NK Cell Therapy	Relapsed or Refractory Acute Myeloid Leukemia	Intravenous infusion	Hangzhou Cheetah Cell Therapeutics Co., Ltd.	Sanhe CHI	Terminated	Unknown	2021	-	NCT05247957
NKG2DL	CAR–NK cells targeting NKG2D ligands	Metastatic Solid Tumors	PBMCs	The Third Affiliated Hospital of Guangzhou Medical University	Guangzhou CHI	Unknown	I	2018	In total, 2 out of 3 patients showed reduced ascites and fewer tumor cells. 1 out of 3 patients experienced rapid tumor regression and complete metabolic liver response.	NCT03415100
NKG2DL	NKG2D CAR–NK	Refractory Metastatic Colorectal Cancer	-	Zhejiang University	Hangzhou CHI	Recruiting	I	2021	-	NCT05213195
NKG2DL	NKX101	Relapsed/Refractory AML and MDS	PB-NK	Nkarta Inc.	Denver CO, Jacksonville FL, Atlanta GA USA	Active, not recruiting	I	2020	Initial Outcomes: NKX101 shows encouraging early responses in relapsed/refractory AML, even in high-risk cases. The toxicity profile matches expectations, with no CRS, ICANS, or treatment-related deaths.	NCT04623944
NKG2DL	NKG2D CAR–NK	AML	Unknown	Zhejiang University	Hangzhou CHI	Recruiting	Unknown	2023	-	NCT05734898
NKG2DL	NKG2D CAR–NK	Ovarian Cancer	Unknown	Hangzhou Cheetah Cell Therapeutics Co., Ltd.	Hangzhou CHI	Recruiting	NA	2023	-	NCT05776355
NKG2DL and PD-L1	PD-1/NKG2D CAR–NK	Non-small Cell Lung Carcinoma	NK-92	Xinxiang medical university	Xinxiang CHI	Completed	I	2018	A previously unreported instance of cytokine release syndrome (CRS) occurred, marking the first known case during CAR–NK therapy	NCT03656705
PD-L1	PD-L1 CAR–NK, Pembrolizumab, N-803	Gastroesophageal Junction (GEJ) Cancers, Advanced HNSCC	t-haNK	National Cancer Institute (NCI)	Bethesda MD USA	Recruiting	II	2021	-	NCT04847466

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
