# Peer review of "Harnessing the Power of NK Cell Receptor Engineering as a New Prospect in Cancer Immunotherapy"

_pharmaceutics, 2024, doi:10.3390/pharmaceutics16091143_

Round 1

Reviewer 1 Report

Comments and Suggestions for Authors

Proposed paper provides an overview of the NK cell receptors that are mainly used as NK cell immunotherapy targets with a focus on the preclinical and clinical trials involving the activating receptors, inhibitory checkpoints, and chemotactic NK cell receptors for monotherapy or combinational approaches in cancer immunotherapy. The review provides a comprehensive understanding of NK cell immunotherapy for tumors. However, some points must be clarified for publication. My detailed suggestions are as follows:

1. The authors aim to introduce the various receptors involved in NK cell immunotherapy for tumors, focusing on those currently in preclinical and clinical research. They provide an extensive introduction and tabulated summary of receptors undergoing clinical research, while offering relatively less information on preclinical studies without summarization. Why? It is recommended to appropriately increase the coverage of preclinical research and ideally summarize it in tabular form to facilitate reading.

2. To enhance the comprehensiveness and appeal of this review, it is recommended to address the current challenges faced by NK cell immunotherapy for tumors. Please discuss the strategies researchers are employing to overcome these obstacles and outline potential future directions for the field. Including these aspects would significantly broaden the audience and impact of the review.

3. It is suggested to further optimize the figures in this article. For instance, use the same color scheme for tumor cells across different figures, and apply consistent colors and shapes for the same receptors to maintain consistency throughout. Additionally, ensure that the legends in Figure 4 are consistent with the illustrations.

4. There are some mistakes in writing in this manuscript. Please check all the manuscript. Such as:

 Both doses tested (1 × 108  and 3 x 108 cells) were well-tolerated, with no signs of neurotoxicity, CRS, or GvHD.

Additionally, the safety and efficacy of the FT538 and daratumumab combination were assessed in another phase 1 clinical study involving AML patients, with doses ranging from 1 x 108  to 15 x 108 cells [clinicaltrials.gov ID: NCT04714372].

Comments on the Quality of English Language

There are some mistakes in writing in this manuscript. Please check all the manuscript. Such as:

 Both doses tested (1 × 108  and 3 x 108 cells) were well-tolerated, with no signs of neurotoxicity, CRS, or GvHD.

Additionally, the safety and efficacy of the FT538 and daratumumab combination were assessed in another phase 1 clinical study involving AML patients, with doses ranging from 1 x 108  to 15 x 108 cells [clinicaltrials.gov ID: NCT04714372].

Author Response

We would like to express our sincere gratitude to reviewer #1 for taking the time to thoroughly read and provide valuable feedback on our manuscript. Taking into consideration reviewer #1’s comments, the manuscript was significantly improved by including a more detailed and balanced conclusion, by giving emphasis to preclinical studies, and by correcting errors throughout the previous version. Below, we address each point that was raised by the reviewer.

Reviewers' comments

Proposed paper provides an overview of the NK cell receptors that are mainly used as NK cell immunotherapy targets with a focus on the preclinical and clinical trials involving the activating receptors, inhibitory checkpoints, and chemotactic NK cell receptors for monotherapy or combinational approaches in cancer immunotherapy. The review provides a comprehensive understanding of NK cell immunotherapy for tumors. However, some points must be clarified for publication. My detailed suggestions are as follows:

  1. The authors aim to introduce the various receptors involved in NK cell immunotherapy for tumors, focusing on those currently in preclinical and clinical research. They provide an extensive introduction and tabulated summary of receptors undergoing clinical research, while offering relatively less information on preclinical studies without summarization. Why? It is recommended to appropriately increase the coverage of preclinical research and ideally summarize it in tabular form to facilitate reading.

Response: We would like to thank the reviewer for the comment. We acknowledge that the previous version of the manuscript lacked a comprehensive summary of the preclinical studies. In the revised manuscript, we have addressed this by summarizing all preclinical studies in a newly created table (Table 2), which includes important details such as the NK cell receptor, the target malignancy or cells, and the key outcomes of each study. We would like to note that, given the extensive number of preclinical studies targeting certain receptors (e.g. CD16 and NKG2D), However, these studies are discussed in the text, with references provided. The new table 2 can be found on page 18, line 754.

  1. To enhance the comprehensiveness and appeal of this review, it is recommended to address the current challenges faced by NK cell immunotherapy for tumors. Please discuss the strategies researchers are employing to overcome these obstacles and outline potential future directions for the field. Including these aspects would significantly broaden the audience and impact of the review.

Response: We agree with the reviewer’s observation that the previous version of the manuscript did not adequately address the current challenges associated with NK cell immunotherapy, nor did it sufficiently highlight how the novel preclinical and clinical studies discussed in this review are attempting to overcome these limitations. In the revised manuscript, we have added new paragraphs to the conclusion section and we have restructured it to present a clearer and more balanced view of the current limitations, including NK cell expansion and activation, tumor infiltration, efficacy in eliminating solid tumors, and the immunosuppressive tumor microenvironment. Furthermore, we have elaborated on the strategies researchers are employing to address each of these limitations, providing examples and discussions of key preclinical and clinical studies. The new paragraphs can be found on page 33, lines 858-943, and are highlighted in purple.

Additionally, we have included two new paragraphs in the conclusion section to highlight our perspective on potential future directions in this field.  The new paragraphs can be found on page 34, line 944-969, and are highlighted in purple.

  1. It is suggested to further optimize the figures in this article. For instance, use the same color scheme for tumor cells across different figures, and apply consistent colors and shapes for the same receptors to maintain consistency throughout. Additionally, ensure that the legends in Figure 4 are consistent with the illustrations.

Response: We would like to thank the reviewer for pointing out the inconsistencies in the figures of the last version of the manuscript. In response to this comment, we have made several adjustments to ensure consistency. Specifically, the color and shape of the receptors in Figures 1, 2, and 4 have been adjusted. Additionally, the color of tumor cells in Figures 1 and 2 has been revised. Finally, the legends in Figure 4 have been modified to align with the illustrations. These adjustments are highlighted in green and can be found on page 31, lines 846-852.

  1. There are some mistakes in writing in this manuscript. Please check all the manuscript. Such as:

 Both doses tested (1 × 108  and 3 x 108 cells) were well-tolerated, with no signs of neurotoxicity, CRS, or GvHD.

Additionally, the safety and efficacy of the FT538 and daratumumab combination were assessed in another phase 1 clinical study involving AML patients, with doses ranging from 1 x 108  to 15 x 108 cells [clinicaltrials.gov ID: NCT04714372].

Response: We appreciate the reviewer for bringing these errors to our attention. We have carefully corrected the identified mistakes, as well as additional mistakes we found in the previous version of the manuscript. These corrections include adjustments to superscripts, commas, and hyphenations, and other grammar errors. The corrections are highlighted in yellow throughout the manuscript.

Reviewer 2 Report

Comments and Suggestions for Authors

The review article provides a comprehensive overview of NK cell receptor engineering as a promising approach in cancer immunotherapy. However, there are several areas where the manuscript could be improved to enhance its clarity, scientific rigor, and overall impact.

1. Clarify and Expand Key Concepts:

Receptor Engineering Strategies: The manuscript touches on various receptor engineering strategies, but it would be helpful to delve deeper into how these techniques specifically enhance NK cell functions. Explaining the underlying mechanisms and how these modifications can lead to better anti-tumor responses would make this section more robust.

Comparison with CAT-T Cell Therapies: While the manuscript correctly highlights the advantages of NK cells over T cells, a brief comparison of NK cell-based therapies with CAR-T cell therapies, particularly in the context of different tumor types (e.g., hematologic vs. solid tumors), would provide valuable context for readers.

2. Incorporate updates on Clinical Trials:

It would also be beneficial to discuss the results of ongoing trials, including patient outcomes and any challenges encountered. Please also comment on the year when each phase of the trials started and if available, the details of any multi-country trials. This could provide a balanced view of where NK cell therapies stand today, including both their potential and limitations.

3. Enhance Structure and Formatting:

Figures and Tables: The inclusion of figures and tables is a strong point, but some, like Figure 1 and 2, could be made clearer with more detailed labels or annotations. Also, adding more specific information to Table 1 and Table 2, such as details about the phases of the clinical trials, patient demographics, year and outcomes, would make these sections more informative.

Consistency in Terminology: To avoid any confusion, it would help to ensure that terminology is consistent throughout the manuscript. Terms like "immune checkpoints," "inhibitory receptors," and "tumor microenvironment" should be clearly defined and used uniformly.

4. Discuss Potential Biases and Limitations:

Balanced Perspective: The manuscript presents a very optimistic view of NK cell receptor engineering, which is great, but it could also benefit from a discussion of potential challenges, such as the difficulty in maintaining NK cell persistence in the body and the risks associated with genetic modifications. Including these points would offer a more rounded perspective.

Ethical Considerations: As genetic engineering rapidly evolves, it’s important to touch on the ethical implications and potential risks of modifying NK cell receptors for clinical use.

Detailed perspective: The conclusions should be even more detailed with a clear future way ahead of using NK cells. The abstract can also be slightly improved with a couple of lines about these perspectives that are to be appeared in the paper.

5. Minor errors:

Presence of highlight on page 7, line 232

There is inconsistent use of hyphenation in phrases such as "cell-based immunotherapies" and "cell engineering." For consistency, decide on whether to hyphenate compound adjectives (e.g., "cell-based" vs. "cell based") and apply it consistently throughout the document.

The citations use different styles for the journal volume and page numbers. For example, some citations use "2020, 17, 807–821" while others use "2021, 12, 350." It would be more consistent to follow one style throughout.

Comments on the Quality of English Language

The quality of English in the manuscript is generally strong, with clear and precise scientific language. However, some sentences are complex and could be simplified for better readability. There are minor issues with punctuation, such as inconsistent comma usage, and some areas where grammar and verb agreement could be refined. Overall, the manuscript maintains a formal tone and logical flow, but minor adjustments could further enhance clarity and readability.

Author Response

We would like to sincerely thank reviewer #2 for their thoughtful and constructive feedback on the manuscript. We believe that the revisions made in response to these suggestions have significantly improved the quality of the manuscript, as well as the clarity, scientific rigor, and overall impact. Below we address each point that was raised by the reviewer.

The review article provides a comprehensive overview of NK cell receptor engineering as a promising approach in cancer immunotherapy. However, there are several areas where the manuscript could be improved to enhance its clarity, scientific rigor, and overall impact.

  1. Clarify and Expand Key Concepts:

Receptor Engineering Strategies: The manuscript touches on various receptor engineering strategies, but it would be helpful to delve deeper into how these techniques specifically enhance NK cell functions. Explaining the underlying mechanisms and how these modifications can lead to better anti-tumor responses would make this section more robust.

Response: We agree with the reviewer’s observation that the manuscript previously lacked sufficient insights into how these techniques can enhance NK cell functions. To address this, we have included new paragraphs to the conclusion section, where we have elaborated on the mechanisms employed by the preclinical and clinical studies to overcome challenges associated with NK cell therapeutics (as noted in comment 4). The new paragraphs are highlighted in purple and can be found on page 33, lines 867-943.

Comparison with CAT-T Cell Therapies: While the manuscript correctly highlights the advantages of NK cells over T cells, a brief comparison of NK cell-based therapies with CAR-T cell therapies, particularly in the context of different tumor types (e.g., hematologic vs. solid tumors), would provide valuable context for readers.

Response: As the reviewer is pointing out, the previous version of the manuscript, while emphasizing the advantages of NK cells over T cells, lacked a comprehensive comparison between CAR-NK and CAR-T cells. To address this, we have added a new paragraph to the CAR-NK section that directly compares CAR-NK to CAR-T cells, with a focus on safety and efficacy across different tumor types. This addition is highlighted in green and can be found on page 30, lines 777-789. Furthermore, the existing text has been reorganized as follows: the explanation of CAR and CD19-CAR-T cells has been moved to the first paragraph, and the discussion of CAR-NK targets for hematological malignancies and solid tumors now follows the new paragraph, continuing the diverse tumor recognition potential of CAR-NK cells. The remainder of the section has been retained in its original order.

  1. Incorporate updates on Clinical Trials:

It would also be beneficial to discuss the results of ongoing trials, including patient outcomes and any challenges encountered. Please also comment on the year when each phase of the trials started and if available, the details of any multi-country trials. This could provide a balanced view of where NK cell therapies stand today, including both their potential and limitations.

Response: We would like to thank the reviewer for pointing out key information that was missing from Table 3 (previously Table 2) and Table 4 (previously Table 3). In the revised manuscript, we have added three new columns to both tables, which now include details on clinical trials involving receptors and clinical trials in combination with CAR-NK cells. These columns provide information on the initiation year of each study, the key outcomes, and study location. Additionally, we have incorporated and discussed some of the key outcomes of these clinical studies in the revised conclusion section, which is highlighted in purple and can be found on page 33.

  1. Enhance Structure and Formatting:

Figures and Tables: The inclusion of figures and tables is a strong point, but some, like Figure 1 and 2, could be made clearer with more detailed labels or annotations. Also, adding more specific information to Table 1 and Table 2, such as details about the phases of the clinical trials, patient demographics, year and outcomes, would make these sections more informative.

Response: As the reviewer correctly indicated, certain figures and tables in the previous version of the manuscript could benefit from enhanced clarity and informativeness. In response, we have made the following improvements in the revised manuscript:

  • In Figure 1, we have added a more detailed explanation of the engineering approaches. Given the already crowded nature of this figure, we refrained from including additional information within the illustration itself. However, the legend of Figure 1 has been expanded, with a more comprehensive explanation of each strategy. The updated legend is highlighted in green and can be found on page 4, lines 154-164.
  • Adjustments have been made to the illustration of Figure 2, including clearer indications of the intracellular and extracellular regions of NK cells and tumor cells, as well as arrow indicators for each enzyme-mediated phosphorylation event. Furthermore, the legend of Figure 2 has been slightly modified and expanded, and it is highlighted in green on page 6, lines 187-195.
  • As mentioned in our response to comment 2, Tables 3 and 4 (previously Tables 2 and 3) have been enriched with three additional columns, which include details on the year, demographics, and key outcomes of each clinical study. The updated Tables 3 and 4 can be found on pages 23 and 32, respectively.

Consistency in Terminology: To avoid any confusion, it would help to ensure that terminology is consistent throughout the manuscript. Terms like "immune checkpoints," "inhibitory receptors," and "tumor microenvironment" should be clearly defined and used uniformly.

Response: We would like to thank the reviewer for identifying inconsistencies in terminology that could potentially lead to confusion. In the revised manuscript, we have addressed this by clearly defining each of these terms in the introduction section. These corrections are highlighted in green and can be found on page 2, lines 75-78, and page 3, lines 121-122. Additionally, we have corrected inconsistencies such as “checkpoint inhibitors” throughout the manuscript. Finally, the acronym of tumor microenvironment (TME) has been defined in the Introduction (page 3, lines 121-122) and consistently used throughout the manuscript.

  1. Discuss Potential Biases and Limitations:

Balanced Perspective: The manuscript presents a very optimistic view of NK cell receptor engineering, which is great, but it could also benefit from a discussion of potential challenges, such as the difficulty in maintaining NK cell persistence in the body and the risks associated with genetic modifications. Including these points would offer a more rounded perspective.

Ethical Considerations: As genetic engineering rapidly evolves, it’s important to touch on the ethical implications and potential risks of modifying NK cell receptors for clinical use.

Detailed perspective: The conclusions should be even more detailed with a clear future way ahead of using NK cells. The abstract can also be slightly improved with a couple of lines about these perspectives that are to be appeared in the paper.

Response: We appreciate reviewer’s insightful comment and we fully agree with these observations. Indeed, the previous version of the manuscript did not sufficiently address the current limitations of NK cell therapies, nor did it provide adequate insights into ethical considerations and future directions. To address this, we have revised the conclusion section to present a more balanced perspective, clearly demonstrating the challenges associated with NK cell therapies. We have also expanded the discussion on these limitations by including examples of relevant preclinical and clinical studies that contribute to the field, along with the mechanisms that they employ. These revisions are highlighted in purple and can be found on page 33-34, lines 858-943.

Additionally, we have included two paragraphs addressing the ethical considerations that should be taken into account for NK cell therapies and NK cell receptor engineering. These additions are also highlighted in purple and can be found on page 35, lines 970-990. Finally, we have added two additional paragraphs in the conclusion section that explore potential future directions for overcoming these challenges. The new paragraphs are highlighted in purple and can be found on page 34, lines 944-969. Finally, the abstract was slightly modified so that the challenges and potential solutions are clearly stated.

We strongly believe that by addressing comment 4, our conclusion section has been significantly improved, offering a more comprehensive and balanced view of NK cell therapeutics and receptor engineering.

  1. Minor errors:

Presence of highlight on page 7, line 232
Response: The mistakes in like 232 were corrected and are highlighted in yellow.

There is inconsistent use of hyphenation in phrases such as "cell-based immunotherapies" and "cell engineering." For consistency, decide on whether to hyphenate compound adjectives (e.g., "cell-based" vs. "cell based") and apply it consistently throughout the document.

Response: We agree with the reviewer’s observation regarding the inconsistent use of hyphenation in the manuscript. In this revised version, we have corrected these inconsistencies, and the adjustments are highlighted in yellow throughout the manuscript.

The citations use different styles for the journal volume and page numbers. For example, some citations use "2020, 17, 807–821" while others use "2021, 12, 350." It would be more consistent to follow one style throughout.

Response: We would like to sincerely thank the reviewer for highlighting mistakes in the references. After a thorough review of the bibliography, we determined that certain references do not provide page range. Instead, they are referring to the article number. Thus, those references have not been altered. However, during our revision, we identified references that were missing entirely page or article numbers, or digital object identifiers (DOI). These references were corrected in the revised version of the manuscript.

Comments on the Quality of English Language

The quality of English in the manuscript is generally strong, with clear and precise scientific language. However, some sentences are complex and could be simplified for better readability. There are minor issues with punctuation, such as inconsistent comma usage, and some areas where grammar and verb agreement could be refined. Overall, the manuscript maintains a formal tone and logical flow, but minor adjustments could further enhance clarity and readability.

Response: In response to the reviewer’s comment, we have corrected grammatical errors, including issues with commas and hyphenation. Additionally, we have refined several sentences to enhance readability. These revisions are highlighted in yellow throughout the revised manuscript.

Round 2

Reviewer 1 Report

Comments and Suggestions for Authors

This manuscript is advised to be accepted after modifying the legends in Figure 4.

Comments on the Quality of English Language

 The mistakes in writing found in this manuscript have all been corrected.

Author Response

We are grateful to Reviewer 1 for the appreciation of our revised manuscript.

We also welcomed the suggestion to improve figure 4 legend. As highlighted in green, in the new revised manuscript we clarified the  text of figure 4 legend and improved the consistency of the color code used for the figure objects.

Reviewer 2 Report

Comments and Suggestions for Authors

The revised manuscript is acceptable for publication, as it has satisfactorily addressed all of my previous queries and concerns. The authors have made significant improvements to the content, ensuring it is well-organized, clear, and scientifically rigorous. The manuscript has also been thoroughly edited for clarity and readability. Given these enhancements, I find the manuscript suitable for publication in its current form.

Author Response

We are grateful to Reviewer 2 for the approval of our revised manuscirpt.